# Integrative small and long RNA omics analysis of human healing and nonhealing wounds discovers cooperating microRNAs as therapeutic targets

Zhuang Liu[1†], Letian Zhang[1†], Maria A Toma[1], Dongqing Li[1,2], Xiaowei Bian[1], Irena Pastar[3], Marjana Tomic-Canic[3], Pehr Sommar[4*], Ning Xu Landén[1,5*]

[1]Dermatology and Venereology Division, Department of Medicine Solna, Center for Molecular Medicine, Karolinska Institutet, Stockholm, Sweden; [2]Key Laboratory of Basic and Translational Research on Immune-Mediated Skin Diseases, Chinese Academy of Medical Sciences; Jiangsu Key Laboratory of Molecular Biology for Skin Diseases and STIs, Institute of Dermatology, Chinese Academy of Medical Sciences and Peking Union Medical College, Nanjing, China; [3]Wound Healing and Regenerative Medicine Research Program, Dr Phillip Frost Department of Dermatology and Cutaneous Surgery, University of Miami Miller School of Medicine, Miami, United States; [4]Department of Plastic and Reconstructive Surgery, Karolinska University Hospital, Stockholm, Sweden; [5]Ming Wai Lau Centre for Reparative Medicine, Stockholm Node, Karolinska Institute, Stockholm, Sweden

**\*For correspondence:**
pehr.sommar@regionstockholm.
se (PS);
ning.xu@ki.se (NXL)

[†]These authors contributed equally to this work

**Abstract** MicroRNAs (miR), as important epigenetic control factors, reportedly regulate wound repair. However, our insufficient knowledge of clinically relevant miRs hinders their potential therapeutic use. For this, we performed paired small and long RNA-sequencing and integrative omics analysis in human tissue samples, including matched skin and acute wounds collected at each healing stage and chronic nonhealing venous ulcers (VUs). On the basis of the findings, we developed a compendium (https://www.xulandenlab.com/humanwounds-mirna-mrna), which will be an open, comprehensive resource to broadly aid wound healing research. With this first clinical, wound-centric resource of miRs and mRNAs, we identified 17 pathologically relevant miRs that exhibited abnormal VU expression and displayed their targets enriched explicitly in the VU gene signature. Intermeshing regulatory networks controlled by these miRs revealed their high cooperativity in contributing to chronic wound pathology characterized by persistent inflammation and proliferative phase initiation failure. Furthermore, we demonstrated that miR-34a, miR-424, and miR-516, upregulated in VU, cooperatively suppressed keratinocyte migration and growth while promoting inflammatory response. By combining miR expression patterns with their specific target gene expression context, we identified miRs highly relevant to VU pathology. Our study opens the possibility of developing innovative wound treatment that targets pathologically relevant cooperating miRs to attain higher therapeutic efficacy and specificity.

## Editor's evaluation

A well-performed study looking at the comprehensive coding and non-coding RNA landscape of the healing wound in a highly controlled fashion. This study lends new insight into specific microRNAs as potential targets in human wound healing.

## Introduction

Wound healing is a fundamental biological process (BP) comprising three sequential and overlapping phases, that is, inflammation, proliferation, and remodeling (*Reinke and Sorg, 2012*). This delicate repair process is often disrupted in chronic venous insufficiency patients, resulting in venous ulcers (VUs) characterized by persistent inflammation and proliferative phase initiation failure (*Eming et al., 2014*). VU is the most common chronic nonhealing wound type, comprising 45–60% of all lower extremity ulcerations (*Vivas et al., 2016*). VU exhibits a marked impact on health-related life quality and represents a significant financial burden both to the patients and the society with an annual health care cost of overall $14.9 billion in the USA (*Hoversten et al., 2020*). A deeper understanding of the underlying gene expression regulatory mechanisms during physiological and pathological wound repair is essential for developing more effective wound treatments (*Stone et al., 2017*).

MicroRNAs (miR) represent a group of short (~22 nt) noncoding (nc) ribonucleic acids, incorporating into the RNA-induced silencing complex and binding to the 3′ untranslated region of their target mRNAs, resulting in mRNA destabilization and translational repression (*Stavast and Erkeland, 2019*). Given that an individual miR can target dozens to hundreds of genes, miRs have been identified as regulators of complex gene networks (*Stavast and Erkeland, 2019*). MiR-mediated regulation is reportedly crucial in multiple fundamental BPs including skin wound repair (*Herter and Xu Landén, 2017*, *Meng et al., 2018*). Importantly, manipulating miRs critical for the disease pathogenesis could offer a prominent therapeutic effect, supported by viral infection- and cancer-targeting miR therapeutics clinical trials (*Rupaimoole and Slack, 2017*). Therefore, miR-based therapeutics for hard-to-heal wounds represent a promising approach (*Herter and Xu Landén, 2017*, *Luan et al., 2018*, *Meng et al., 2018*; *Nie et al., 2020*; *Pastar et al., 2020*, *Sen and Roy, 2012*).

However, our insufficient knowledge of the miR-mediated gene regulation in human wounds severely hinders the identification of clinically relevant miRs and their potential therapeutic use. While most previous wound healing-related miR studies rely on in vitro or animal models, only a few have approached miR profiles in human wound tissues or primary cells from patients, including tissues and fibroblasts of diabetic foot ulcers (*Liang et al., 2016*; *Ramirez et al., 2018*), burn wound dermis (*Liang et al., 2012*), and acute wounds at the inflammatory phase (*Li et al., 2015*). Despite sharing several fundamental features, the human skin structure and repair processes are different from those of the commonly used animal models (e.g., rodents) (*Elliot et al., 2018*). Moreover, animal models cannot fully simulate the human disease complexity, and the findings are difficult to extrapolate to humans (*Darwin and Tomic-Canic, 2018*; *Pastar et al., 2018*). Thereby, a rigorous and in-depth characterization of miR-mediated gene regulatory networks in human healing and nonhealing wounds is timely needed.

In this study, we performed paired small and mRNA expression profiling in the human skin, acute wounds during the inflammatory and proliferative phases, and VU, unraveling time-resolved changes of the whole transcriptome throughout the wound healing process and the unique gene expression signature of a common chronic wound type. The integrative miR and mRNA omics analysis provides a network view of miR-mediated gene regulation in human wounds in vivo and demonstrates the functional involvement of miRs in human skin wound repair at the system level. Importantly, we identified miRs highly relevant to VU pathology, based not only on their aberrant expression but also their targetome enriched in the VU-related gene expression signature. Apart from confirming the in silico findings, the experimental miR expression, targetome, and function validation uncovered that VU-dysregulated miRs could act cooperatively contributing to the stalled wound healing characterized by failed transition from inflammatory-to-proliferative phase, which opens up new possibility for the development of more precise and innovative wound treatment targeting pathologically relevant cooperating miRs to achieve higher therapeutic efficacy and specificity. Additionally, based on this comprehensive analysis of human wound tissues, we built a browsable resource web portal (https://www.xulandenlab.com/humanwounds-mirna-mrna), which is the first wound healing-focused miR resource for facilitating the exploration of miR's clinical application and for aiding in the elucidation of posttranscriptional regulatory underpinnings of tissue repair.

## Results

### miRNA and mRNA paired expression profiling in human wounds

To better understand tissue repair in humans, we collected wound-edge tissues from human acute wounds and chronic nonhealing VUs (*Figure 1a* and Materials and methods, Tables 2 and 3). Donor demographics are presented in *Table 1*. We created 4-mm full-thickness punch wounds at the lower legs of healthy volunteers aged beyond 60 years to match the advanced age of VU patients and anatomical location of the highest VUs occurrence (*Vivas et al., 2016*). Tissue was collected at baseline (Skin), and at day 1 and 7 post-wounding (Wound1 and Wound7) to capture the inflammatory and proliferative phases of wound healing, respectively. In total, 20 samples divided into four groups, that is, Skin, Wound1, Wound7, and VU, were analyzed by Illumina small RNA sequencing (sRNA-seq) and ribosomal RNA-depleted long RNA sequencing (RNA-seq). After stringent raw sequencing data quality control (*Supplementary file 1* and *Supplementary file 2*), we detected 562 mature miRs and 12,069 mRNAs in our samples. Our principal component analysis showed that either the miR or the mRNA expression profiles clearly separated these four sample groups (*Figure 1b*). Next, we performed pairwise comparisons to identify the differentially expressed genes (DEGs) during wound repair. We compared the VUs with both the skin and acute wounds and unraveled a VU-specific gene signature, including aberrant increase of 22 miRs and 221 mRNAs and decrease of 10 miRs and 203 mRNAs (differentially expressed [DE] analysis false discovery rate [FDR] <0.05, fold change ≥2 for miRs, and ≥1.5 for mRNAs, *Figure 1c–e* and *Figure 1—source data 1*). The full DEG list can be browsed on our resource website (https://www.xulandenlab.com/humanwounds-mirna-mrna) with more or less rigorous cutoffs. With this unique resource, we dissected further the miR-mediated posttranscriptional regulatory underpinnings of wound repair.

### Dynamically changed miR expression during wound repair

We leveraged weighted gene coexpression network analysis (WGCNA) for classifying miRs according to their coexpression patterns in the 20 sRNA-seq analyzed samples to link the miR expression changes with wound healing progression or nonhealing status at a system level (*Langfelder and Horvath, 2008*). We identified 13 distinct modules with a robustness confirmed by the module preservation analysis (*Figure 2—figure supplement 1a*), 10 of them significantly correlating (Pearson's correlation, FDR <0.05) with at least one of the four phenotypic traits, that is, Skin, Wound1, Wound7, and VU (*Figure 2a, b*, *Figure 2—source data 1*). The WGCNA revealed that module (m)2, m10, and m11 miRs were upregulated at the inflammatory phase (Wound1), while m5 and m6 miRs peaked at the proliferative phase (Wound7). In VU, we identified three downregulated (m3, m7, and m9) and two upregulated (m8 and m12) miR modules. We highlighted the 198 'driver' miRs (i.e., the top 20 miRNAs with the highest kME values in each module and kME >0.5) of the 10 significant modules in the coexpression networks (*Figure 2c, d* and *Figure 2—figure supplement 2b–i*) and they could also be browsed on our resource web portal (https://www.xulandenlab.com/humanwounds-mirna-mrna). Notably, we identified 84% of them as DEGs, suggesting a high consistence between the WGCNA and DE analysis (*Figure 2—source data 2*).

We hypothesized that the coexpression of various miRs could be due to their transcription driven by common transcription factors (TFs). To test this idea, we leveraged TransmiR v2.0 (*Tong et al., 2019*), a database including literature-curated and ChIP-seq-derived TF-miR regulation data, to identify the enriched TFs in each miR module (Fisher's extract test: odds ratio [OR] >1, FDR <0.05, *Figure 2—source data 3*). Interestingly, the KLF4, KLF5, GATA3, GRHL2, and TP53 families exhibited not only their binding sites enriched in the m9 miR genes but their expression also significantly correlated with the m9 miRs (Pearson's correlation coefficient = 0.53–0.82, p value of p = 7.05e−06–0.014) (*Figure 2d*, *Figure 2—figure supplement 3a*, and *Figure 2—source data 3*). Notably, both GATA3 and KLF4 are reportedly downregulated in human VU (*Stojadinovic et al., 2014*; *Stojadinovic et al., 2008*), which findings were confirmed in our RNA-seq data (*Figure 2—figure supplement 3b*). These results could explain the deficiency of their regulated miRs in VU and also suggest a link between these TFs and chronic wound pathology.

### mRNA coexpression networks underpinning wound repair

miRs exert functions through the posttranscriptional regulation of their target mRNAs. Therefore, describing the mRNA expression context would be required for understanding the role of miRs

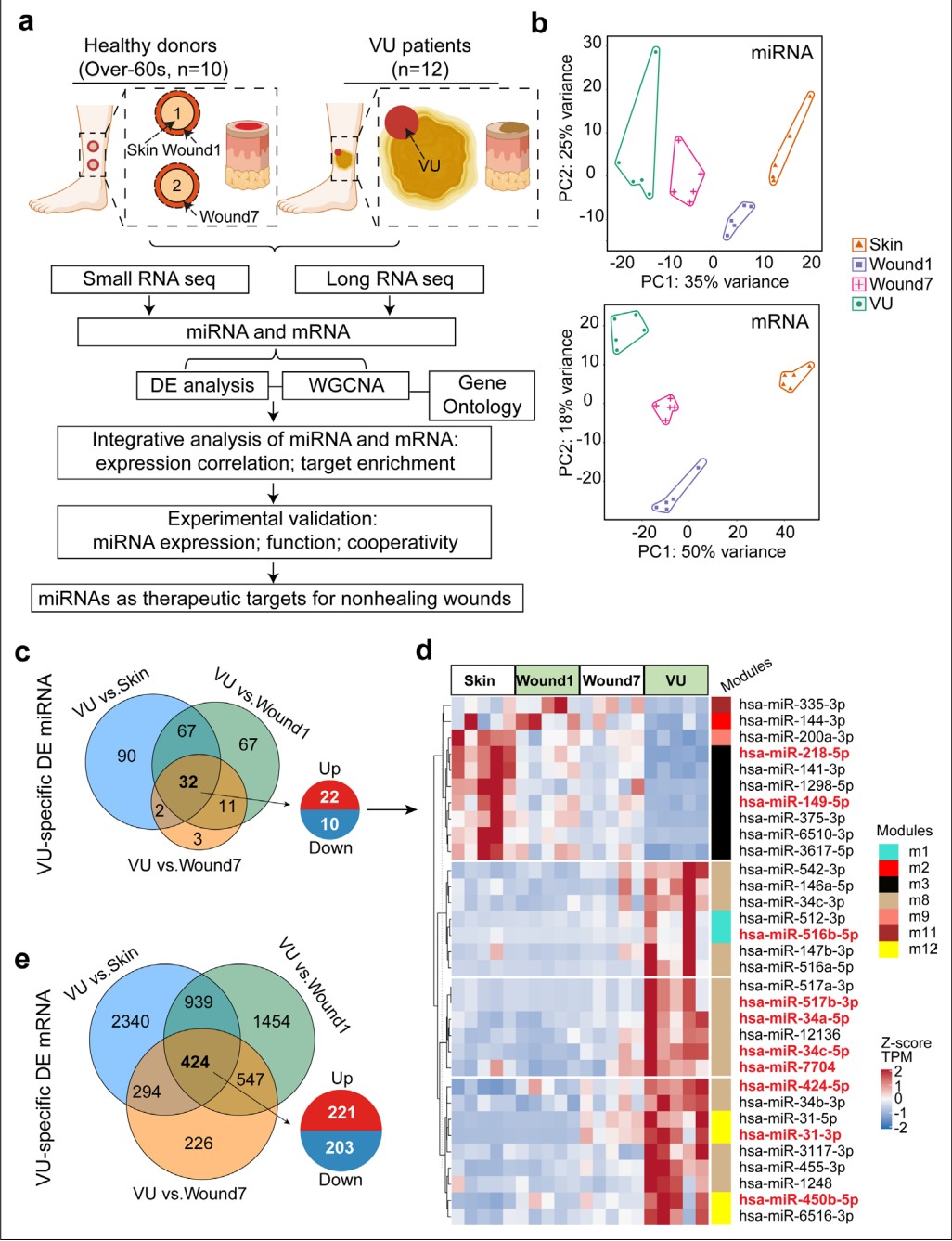

**Figure 1.** Paired profiling of miRNA and mRNA expression in human wounds. (**a**) Schematic of analysis in this study (*n* = samples used for RNA-seq and qRT-PCR validation in full-thickness tissues). (**b**) Principal component analysis (PCA) plots based on miRNA (upper panel) and mRNA (lower panel) expression profiles. Each dot indicates an individual sample. The numbers of differentially expressed (DE) miRNAs (**c**) and mRNAs (**e**) in venous ulcer (VU) (*n* = 5) compared to the Skin, Wound1, and Wound7 from five healthy donors are shown in Venn diagrams. False discovery rate (FDR) <0.05, fold change ≥2 for miRNAs, and ≥1.5 for mRNAs. (**d**) The heatmap depicts the 32 miRNAs specifically dysregulated in the VU with scaled expression values (Z-scores). Weighted gene coexpression network analysis (WGCNA) modules of each miRNA belongs to are marked with color bars. The miRNAs with experimentally validated expression changes are highlighted in red.

The online version of this article includes the following source data for figure 1:

**Source data 1.** miRNAs and mRNA with expression change specifically in venous ulcers.

**Table 1.** Characteristics of the healthy donors and the venous ulcer patients.

| Characteristics | Healthy donors | Patients with VU |
|---|---|---|
| Study population (*n*) | 10 | 12 |
| Age, years (mean ± SD) | 65.3 ± 3.2 | 75.8 ± 12.0 |
| Ethnicity | Caucasian | Caucasian |
| Gender (male:female) | 2:8 | 5:7 |
| Biopsy location | Lower leg | Lower leg |
| Wound duration | Acute (1 or 7 days after injury) | 3.7 ± 5.3 years |

SD, standard deviation; VU, venous ulcer (*n* = samples used for RNA-seq and qRT-PCR validation).

in wound repair (*Agarwal et al., 2015*). We thus performed WGCNA in the paired long RNA-seq data and identified 13 mRNA coexpression modules (*Figure 2e*, *Figure 2—figure supplement 1b*, *Figure 2—figure supplement 4a–c*, and *Figure 2—source data 4*). The gene ontology (GO) analysis of the mRNA modules largely confirmed the previous knowledge of wound biology, such as skin hemostasis (M2) and barrier function (M4)-related gene downregulation in the wounds, the upregulation of the genes involved in the immune response (M8), RNA processing, and protein production (M1, M3, and M5) in the inflammatory phase, and the prominent cell mitosis-related gene expression (M7) in the proliferative phase of wound repair (*Figure 2f* and *Figure 2—figure supplement 5a*). These results further supported the robustness and reproducibility of our profiling data. Moreover, this unique dataset allows the identification of the key TFs driving these BPs. For example, we identified NFKB1 and RELA, well known for their immune functions (*Liu et al., 2017*), as the most enriched upstream regulators for the M1 mRNAs, while E2F1, a TF promoting cell growth (*Ertosun et al., 2016*), surfaced as a master regulator TF in M7 (*Figure 2—source data 5*).

Importantly, our study unraveled a VU molecular signature: downregulated expression of RNA and protein production- (M1, M3, and M5) as well as cell mitosis-related (M7) genes, and upregulated expression of genes involved in extracellular matrix organization and cell adhesion (M9). These results were in line with the dermal tissue fibrosis observed in patients with chronic venous insufficiency (*Blumberg et al., 2012*; *Pappas et al., 2020*, *Stone et al., 2020*). Moreover, we found an immune gene signature clearly distinguishing the chronic inflammation in VUs (M11 and M12 enriched with adaptive immunity-related mRNAs) from the self-limiting immune response in acute wounds (M8 enriched with neutrophil activation- and phagocytosis-related mRNAs) (*Figure 2f* and *Figure 2—figure supplement 5b*). Overall, we generated a gene expression map of human healing and nonhealing wounds, setting a steppingstone for the in-depth understanding of the VU pathological mechanisms. After having established this map, we decided to dissect how miRs contribute to these pathological changes.

## Integrative analysis of miR and mRNA expression changes in wound healing

Among the multiple gene expression regulatory mechanisms, we aimed to evaluate how miRs could contribute to the protein-coding gene expression in human wound repair. We thus performed a correlation analysis for the miRs and mRNAs that were DE in VU compared to the skin and acute wounds, using the first principal component (PC1) of their expression in each sample. We found significantly negative correlations (Pearson's correlation, p values: 1.36e−12–1.27e−04) between the PC1 of the DE miRs and the DE mRNAs predicted as miR targets, indicating negative regulation of VU-mRNA signature by the aberrantly expressed miRs in VU (*Figure 3a–c*).

Furthermore, we dissected the potential regulatory relationship between the VU-associated miR and mRNA modules. We identified significantly negative correlations between the downregulated miR (m3, m7, and m9) and the upregulated mRNA (M9, M11, and M12) modules, as well as between the upregulated miR (m8 and m12) and the downregulated mRNA (M5) modules in VU (*Figure 3d*). Among these miR–mRNA module pairs, we found that the predicted targets of the downregulated m9 miRs were significantly enriched (Fisher's extract test: OR >1, p value <0.05) for the upregulated M9 mRNAs, whereas the targets of the upregulated miRs and m8 miRs were enriched for the

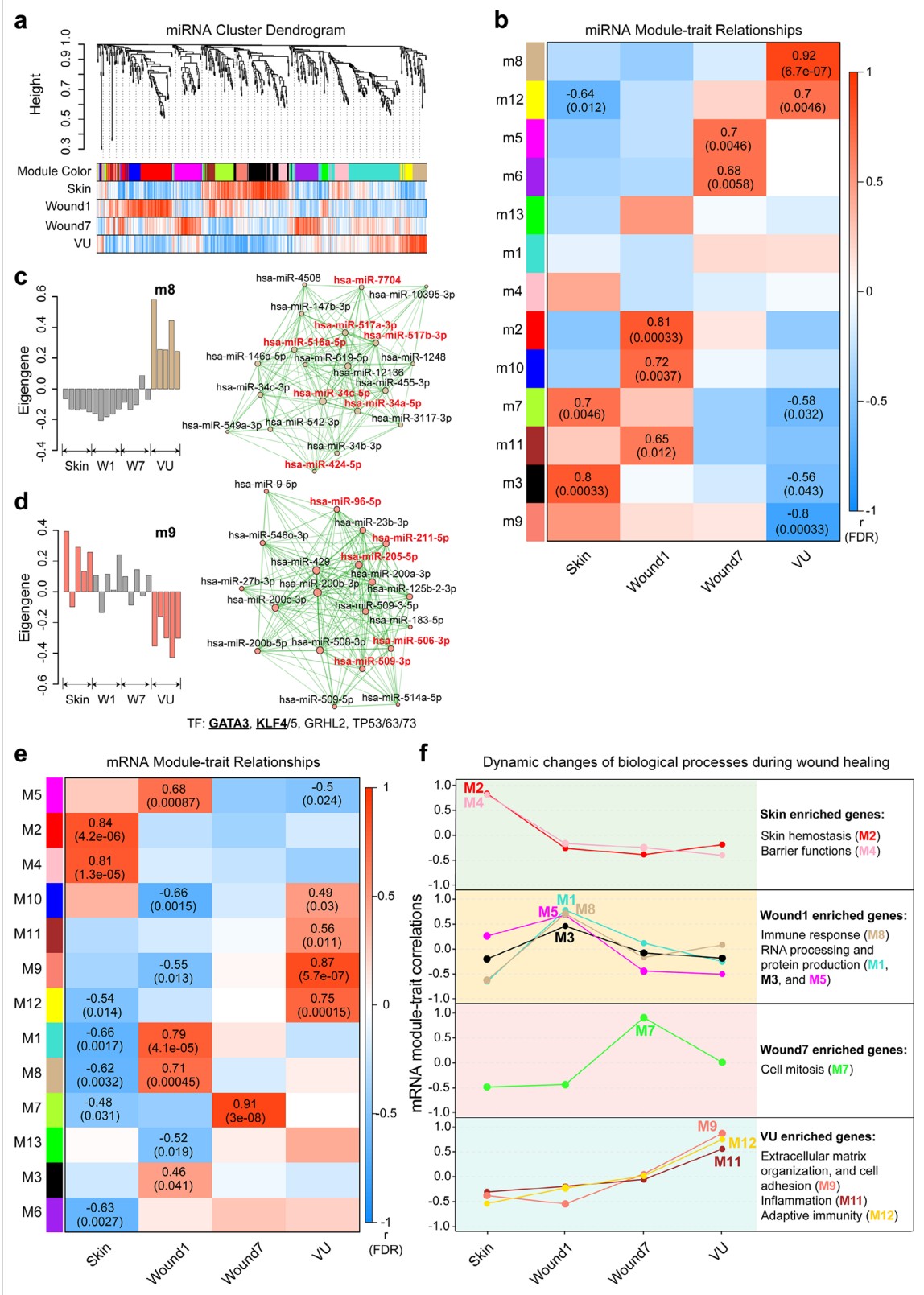

**Figure 2.** Weighted gene coexpression network analysis (WGCNA) of miRNAs and mRNAs in wound healing. (**a**) Cluster dendrogram shows miRNA coexpression modules: each branch corresponds to a module, and each leaf indicates a single miRNA. Color bars below show the module assignment (the first row) and Pearson's correlation coefficients between miRNA expression and the sample groups (the second to the fifth row: red and blue lines represent positive and negative correlations, respectively). (**b**) Heatmap shows Pearson's correlations between miRNA module eigengenes (MEs) and the

*Figure 2 continued on next page*

*Figure 2 continued*

sample groups. The correlation coefficients and the adjusted p values (false discovery rate, FDR) are shown where the FDRs are less than 0.05. For the venous ulcer (VU)-associated modules m8 (**c**) and m9 (**d**), bar plots (left) depict the ME values across the 20 samples analyzed by RNA-seq, and network plots (right) show the top 20 miRNAs with the highest kME values in each module. Node size and edge thickness are proportional to the kME values and the weighted correlations between two connected miRNAs, respectively. The miRs with their targetome enriched with VU-mRNA signature (see *Figure 4b*) are highlighted in red. Transcription factors (TFs) with their targets enriched in the m9 module (Fisher's exact test: false discovery rate [FDR] <0.05) are listed below the network, and TFs differentially expressed in VU are underlined. (**e**) Heatmap shows Pearson's correlations between mRNA MEs and the sample groups. (**f**) The gene expression pattern of each module across all the sample groups is depicted with line charts. Gene ontology analysis of biological processes enriched in each module is shown at the right.

The online version of this article includes the following source data and figure supplement(s) for figure 2:

**Source data 1.** Weighted gene coexpression network analysis of miRNAs in wound healing.

**Source data 2.** Top 20 driver miRNAs of each significant module in weighted gene coexpression network analysis (WGCNA).

**Source data 3.** Transcription factors (TFs) regulating miRNA expression in each module.

**Source data 4.** Weighted gene coexpression network analysis of mRNAs in wound healing.

**Source data 5.** Transcription factors (TFs) with targets enriched in significant mRNA modules.

**Figure supplement 1.** Preservation analysis of miRNA and mRNA coexpression modules.

**Figure supplement 2.** Weighted gene coexpression network analysis (WGCNA) of miRNAs in human skin wound healing (related to *Figure 2a–d*).

**Figure supplement 3.** Transcription factors (TFs) with targets enriched in the miRNA m9 module (related to *Figure 2d*).

**Figure supplement 4.** Weighted gene coexpression network analysis (WGCNA) of mRNAs in human skin wound healing (related to *Figure 2e*).

**Figure supplement 5.** Functional enrichment analysis for mRNA modules (related to *Figure 2f*).

---

downregulated mRNAs and M5 mRNAs (*Figure 4a* and *Figure 4—source data 1*). These results suggest that miRs contribute to the aberrant mRNA expression in VU at a global level.

Based on the above-identified miR–mRNA module pairs, we next searched for individual candidate miRs with their targets enriched for the VU mRNA signature. We observed that the targets of two VU-associated downregulated miRs (miR-144-3p and miR-218-5p) and five m9 miRs (miR-205-5p, miR-211-5p, miR-506-3p, miR-509-3p, and miR-96-5p) were enriched for the upregulated M9 mRNAs, whereas the targets of three VU-associated upregulated miRs (miR-450-5p, miR-512-3p, and miR-516b-5p) and seven m8 miRs (miR-424-5p, miR-34a-5p, miR-34c-5p, miR-516a-5p, miR-517a-3p, miR-517b-3p, and miR-7704) were enriched for M5 mRNAs and downregulated mRNAs (*Figure 4b* and *Figure 4—source data 2*). These miR targetomes were enriched for the mRNAs associated with VU pathology. Therefore, these miR candidates are of importance for understanding the pathological mechanisms hindering wound healing. Moreover, we compiled miR-mediated gene expression regulation networks centered with these highly pathologically relevant miRs (*Figure 4c*, *Figure 4—figure supplements 1 and 2*, and *Figure 4—source data 2*). These networks also include the mRNAs predicted as the strongest targets and with anticorrelated expression patterns with these miRs in human wounds in vivo, as well as the TFs reported to regulate these miR expressions from the TransmiR v2 database (*Tong et al., 2019*). Taken together, our study identifies a list of highly pathological relevant miRs and their targetomes in human VU.

## Experimental validation of miR expression and targets in human skin wounds

We next experimentally validated our in silico findings about the VU relevant miRs, including their expression, targetome, and biological functions (*Figure 5a*). First, we selected 9 shortlisted DE miRs (*Figures 1d and 4b*), including 3 downregulated (miR-149-5p, miR-218-5p, and miR-96-5p) and 6 upregulated (miR-7704, miR-424-5p, miR-31-3p, miR-450-5p, miR-516b-5p, and miR-517b-3p) miRs in VU, and validated their expression by quantitative reverse transcription PCR (qRT-PCR) in the skin and wound tissue biopsies from a cohort with 7 healthy donors and 12 VU patients, matched in terms of age and the anatomical wound locations (*Figure 5b–j*, *Tables 1 and 2*, and *Figure 5—source data 1*). Additionally, the upregulation of miR-34a-5p and miR-34c-5p in VU has been reported by us previously (*Wu et al., 2020*). Together, we confirmed these 11 miRs' expression patterns in RNA-seq, supporting the robustness and reproducibility of our profiling data (*Figure 5—figure supplement 1*). Next, we purified CD45⁻ epidermal cells consisting of mainly keratinocytes from matched human

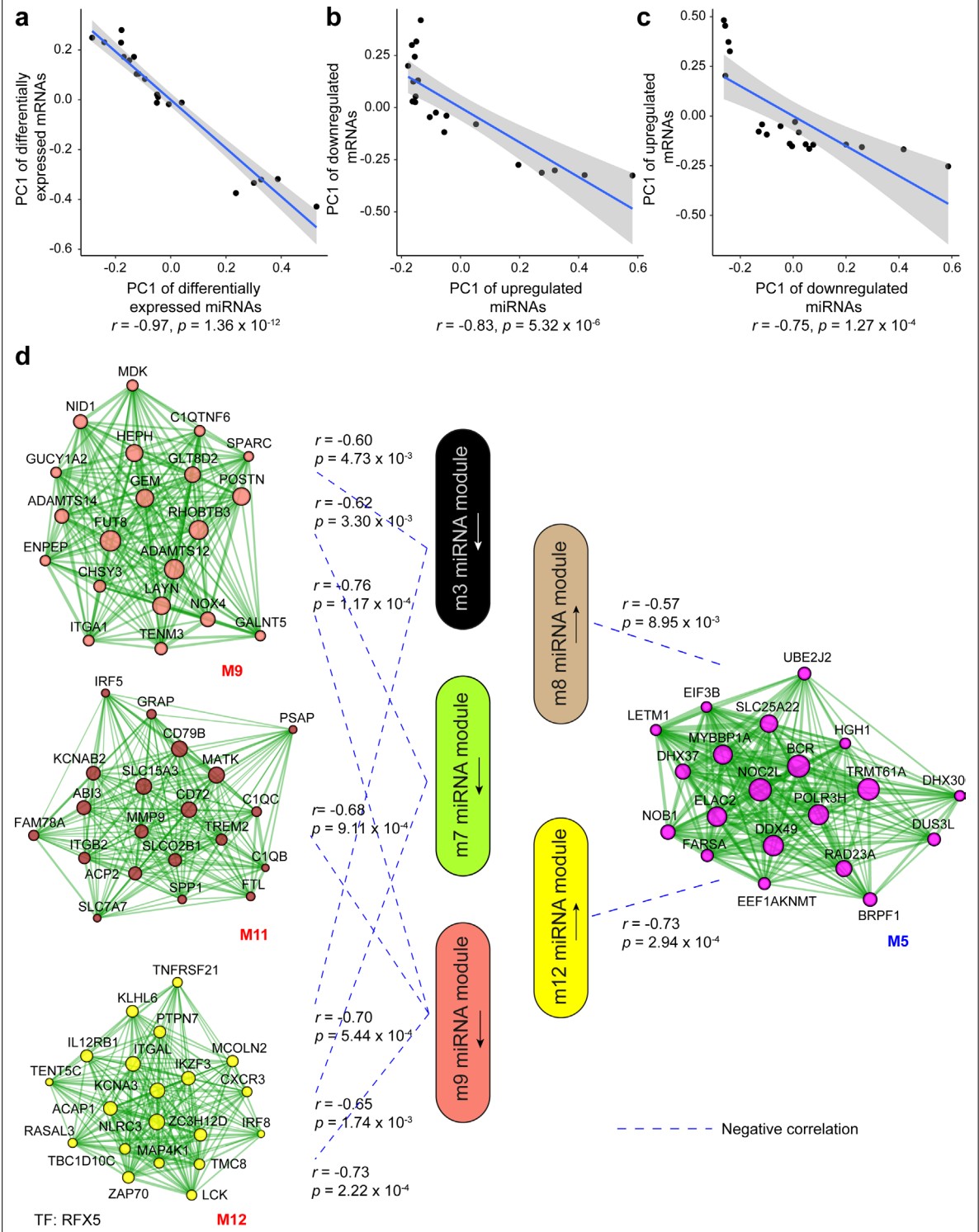

**Figure 3.** Correlation analysis between miRNA and mRNA expression changes in venous ulcer (VU). (**a–c**) Correlations between the first principal component (PC1) of VU-associated differentially expressed (DE) miRNAs, and the PC1 of VU DE mRNAs predicted as miRNA targets. (**d**) PC1 correlations between the hub miRNAs and their predicted targets in the VU-associated miRNA and mRNA modules. Pearson's correlation coefficients (*r*) and p values are shown. The mRNA networks are plotted with the top 20 most connected module genes. Transcription factors (TFs) with targets enriched in the VU-associated modules are listed below the networks.

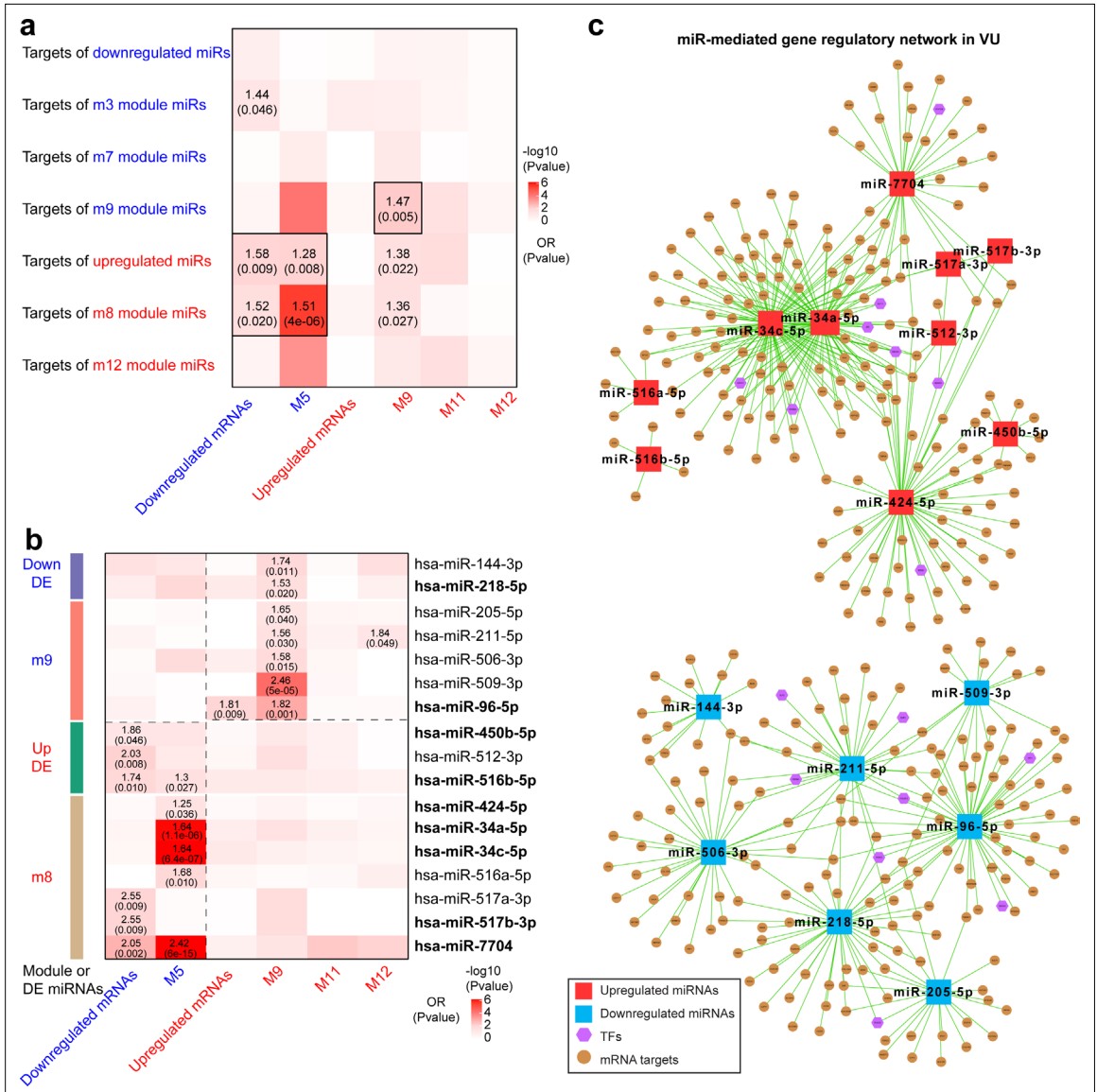

**Figure 4.** Integrative analysis of miRNA and mRNA expression changes in venous ulcer (VU). Heatmaps show the enrichment for VU-affected mRNAs and mRNA modules in the top targets of (**a**) VU-associated differentially expressed (DE) miRNAs and miR modules and (**b**) the individual candidate miRNAs. Odds ratio (OR) and p values are indicated when OR >1 and p value <0.05 (Fisher's exact test). The miRNAs with experimentally validated expression changes are highlighted in bold. (**c**) miR-mediated gene regulatory networks in VU are constructed with the miRNAs in (b), the mRNAs predicted as the strongest targets and with anticorrelated expression patterns (Pearson's correlation, p value <0.05 and $r < 0$) with these miRNAs in human wounds, and the transcription factors (TFs) regulating these miRNAs' expression from the TransmiR v2 database. An enlarged version of these networks can be found in *Figure 4—figure supplements 1 and 2*.

The online version of this article includes the following source data and figure supplement(s) for figure 4:

**Source data 1.** Gene set enrichment analysis for VU-affected DE mRNAs and mRNA modules in the strongest targets of VU-associated DE miRNAs and miRNA modules.

**Source data 2.** Individual candidate miRNAs with their targets enriched for the VU mRNA signature.

**Figure supplement 1.** MiR-mediated gene regulatory network in VU, which is constructed with the upregulated miRNAs in *Figure 4b*, the mRNAs predicted as the strongest targets and with anticorrelated expression patterns (Pearson's correlation, p value <0.05 and $r < 0$) with these miRNAs in human wounds, and the transcription factors (TFs) regulating these miRNAs' expression from the TransmiR v2 database (related to *Figure 4b, c* and *Figure 4—source data 2*).

**Figure supplement 2.** MiR-mediated gene regulatory network in VU, which is constructed with the downregulated miRNAs in *Figure 4b*, the mRNAs predicted as the strongest targets and with anticorrelated expression patterns (Pearson's correlation, p value <0.05 and $r < 0$) with these miRNAs in human wounds, and the transcription factors (TFs) regulating these miRNAs' expression from the TransmiR v2 database (related to *Figure 4b, c* and *Figure 4—source data 2*).

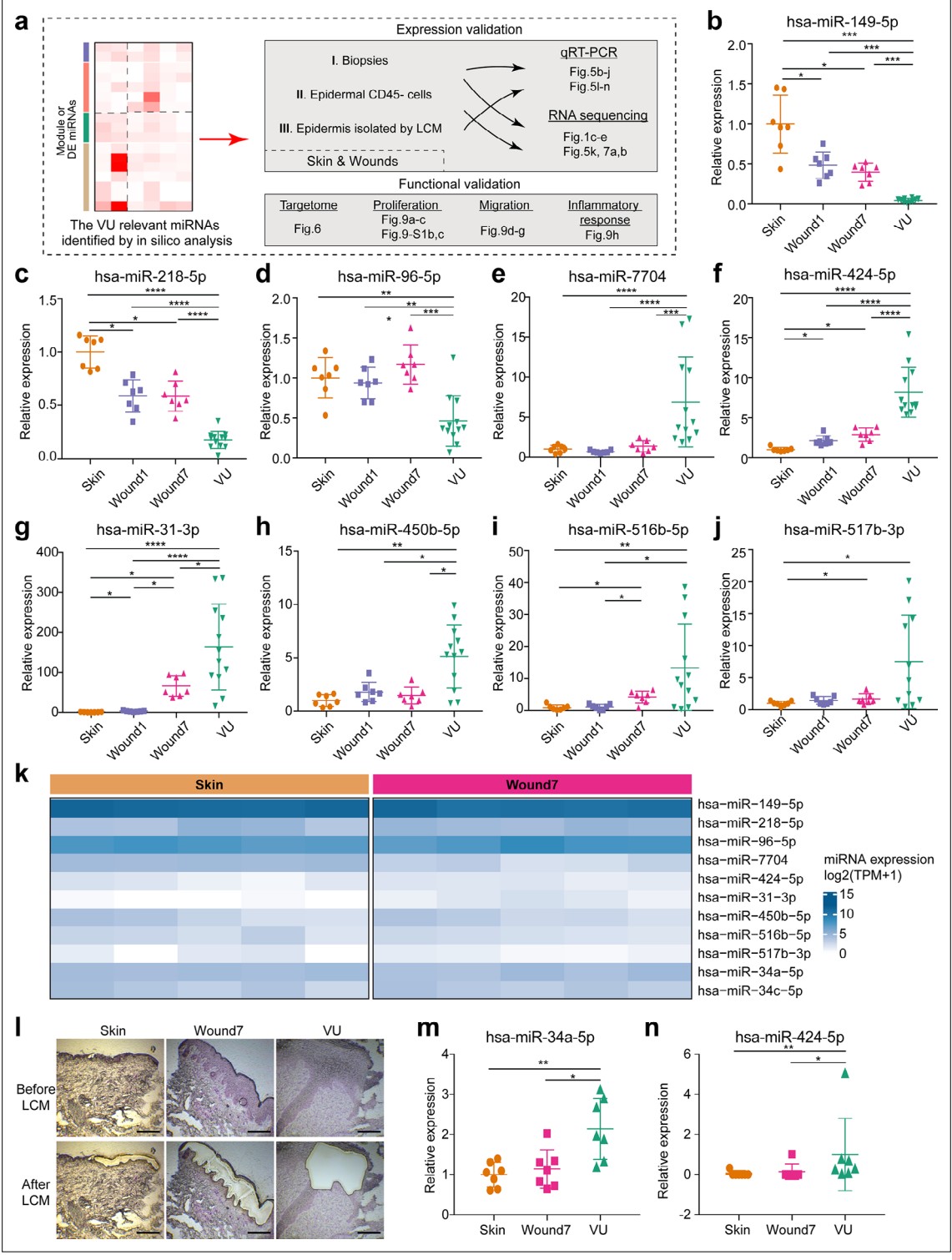

**Figure 5.** Experimental validation of miRNAs' expression in human skin wounds. (**a**) A schematic diagram of experimental validation in this study. (**b–j**) qRT-PCR analysis of venous ulcer (VU)-associated differentially expressed (DE) miRNAs in the skin, day 1 and 7 acute wounds from seven healthy donors and VU from 12 patients. (**k**) VU-associated DE miRNA expression in RNA-seq of epidermal keratinocytes from human skin and wound day 7. (**l**) Representative photographs of laser capture microdissection (LCM) -isolated epidermis from skin, wound7 and VU. qRT-PCR analysis of (**m**) miR-34a-5p and (**n**) miR-424-5p in LCM-isolated epidermis (*n* = 7). The data are presented as mean ± standard deviation (SD). Wilcoxon signed-rank test was used for the comparison between Skin, Wound1, and Wound7; Mann–Whitney *U*-test was used for comparing VU with the skin and acute wounds. *p < 0.05, **p < 0.01, ***p < 0.001, ****p < 0.0001.

*Figure 5 continued on next page*

*Figure 5 continued*

The online version of this article includes the following source data and figure supplement(s) for figure 5:

**Source data 1.** Experimental validation of miRNAs' expression in human skin wounds (related to *Figure 5b–j*), or in LCM-isolated epidermis (related to *Figure 5m, n*), enrichment analysis of the experimentally validated miRNA targets for the venous ulcer (VU) gene signature (related to *Figure 6k*), miRNAs' overexpression efficiency (related to *Figure 6—figure supplement 1*), and miRNA targets validated by the microarray and in VU gene signature (related to *Figure 7a and b*, lower panels).

**Figure supplement 1.** RNA-sequencing results for the miRNAs selected for experimental validation.

skin and acute wounds of five healthy donors. With sRNA-seq, we confirmed the in vivo expression of these 11 miRs in the epidermal cells of human skin and wounds (*Figure 5k*). In addition, with laser capture microdissection (LCM), we isolated the epidermal compartments from human skin, acute wounds, and VUs (*n* = 7 per group) and found that the expression of miR-34a-5p and miR-424-5p was upregulated in the wound-edge epidermis from the VUs compared to the acute wounds and the skin, as shown in qRT-PCR analysis (*Figure 5l–n*). These results provided a rationale for the selection of human primary keratinocytes to further study these miRs' targets and functions.

Furthermore, we experimentally validated the targets of eight miRs surfaced in our analysis (*Figure 4b*), including the miRs downregulated (miR-218-5p and miR-96-5p) and upregulated (miR-424-5p, miR-450-5p, miR-516b-5p, miR-34a-5p, miR-34c-5p, and miR-7704) in VU. We performed genome-wide microarray analysis in human primary keratinocytes or fibroblasts overexpressing each of these miRs (*Figure 6a*, *Figure 6—figure supplement 1*, and *Figure 5—source data 1*). Also, we reanalyzed our published microarray dataset on keratinocytes with miR-34a-5p or miR-34c-5p overexpression (GSE117506) (*Wu et al., 2020*). For all these eight miRs, we observed that their strongest targets predicted by TargetScan were significantly downregulated compared to the nontargeting

**Table 2.** Characteristics of the patients with venous ulcer.

| Patient | Sex | Age (years) | Ethnicity | Wound size (cm) | Wound duration (years) | Wound location | Experiment |
|---|---|---|---|---|---|---|---|
| 1 | M | 86 | Caucasian | 6 × 5 | 4 | Lower leg | RNA-seq and qRT-PCR |
| 2 | F | 68 | Caucasian | 15 × 15 | 2 | Lower leg | RNA-seq and qRT-PCR |
| 3 | M | 70 | Caucasian | 3 × 0.5 | 0.3 | Lower leg | RNA-seq and qRT-PCR |
| 4 | F | 78 | Caucasian | 15 × 12 | 1.5 | Lower leg | RNA-seq and qRT-PCR |
| 5 | F | 87 | Caucasian | 3 × 2 + 8 × 4 | 2.5 | Lower leg | RNA-seq and qRT-PCR |
| 6 | F | 73 | Caucasian | 3 × 4 | 3.5 | Lower leg | qRT-PCR |
| 7 | M | 99 | Caucasian | 20 × 10 | 0.5 | Lower leg | qRT-PCR |
| 8 | F | 71 | Caucasian | 20 × 20 | 20 | Lower leg | qRT-PCR |
| 9 | F | 77 | Caucasian | 2.5 × 3 + 15 × 15 | 4.5 | Lower leg | qRT-PCR |
| 10 | M | 51 | Caucasian | 2 × 1.5 | 3 | Lower leg | qRT-PCR |
| 11 | M | 69 | Caucasian | 12 × 15 | 1 | Lower leg | qRT-PCR |
| 12 | F | 81 | Caucasian | 7 × 2.5 | 1 | Lower leg | qRT-PCR |
| 13 | M | 65 | Caucasian | 3 × 3 | 2.2 | Lower leg | LCM |
| 14 | M | 78 | Caucasian | 10 × 5 | 0.9 | Lower leg | LCM |
| 15 | M | 63 | Caucasian | 4 × 4 | 0.3 | Lower leg | LCM |
| 16 | M | 50 | Caucasian | 4.5 × 1 | 0.4 | Lower leg | LCM |
| 17 | M | 71 | Caucasian | 13 × 10 | 1 | Lower leg | LCM |
| 18 | M | 92 | Caucasian | 12 × 12 | 2 | Lower leg | LCM |
| 19 | F | 87 | Caucasian | 3 × 3 | 2.5 | Lower leg | LCM |

M, male; F, female.

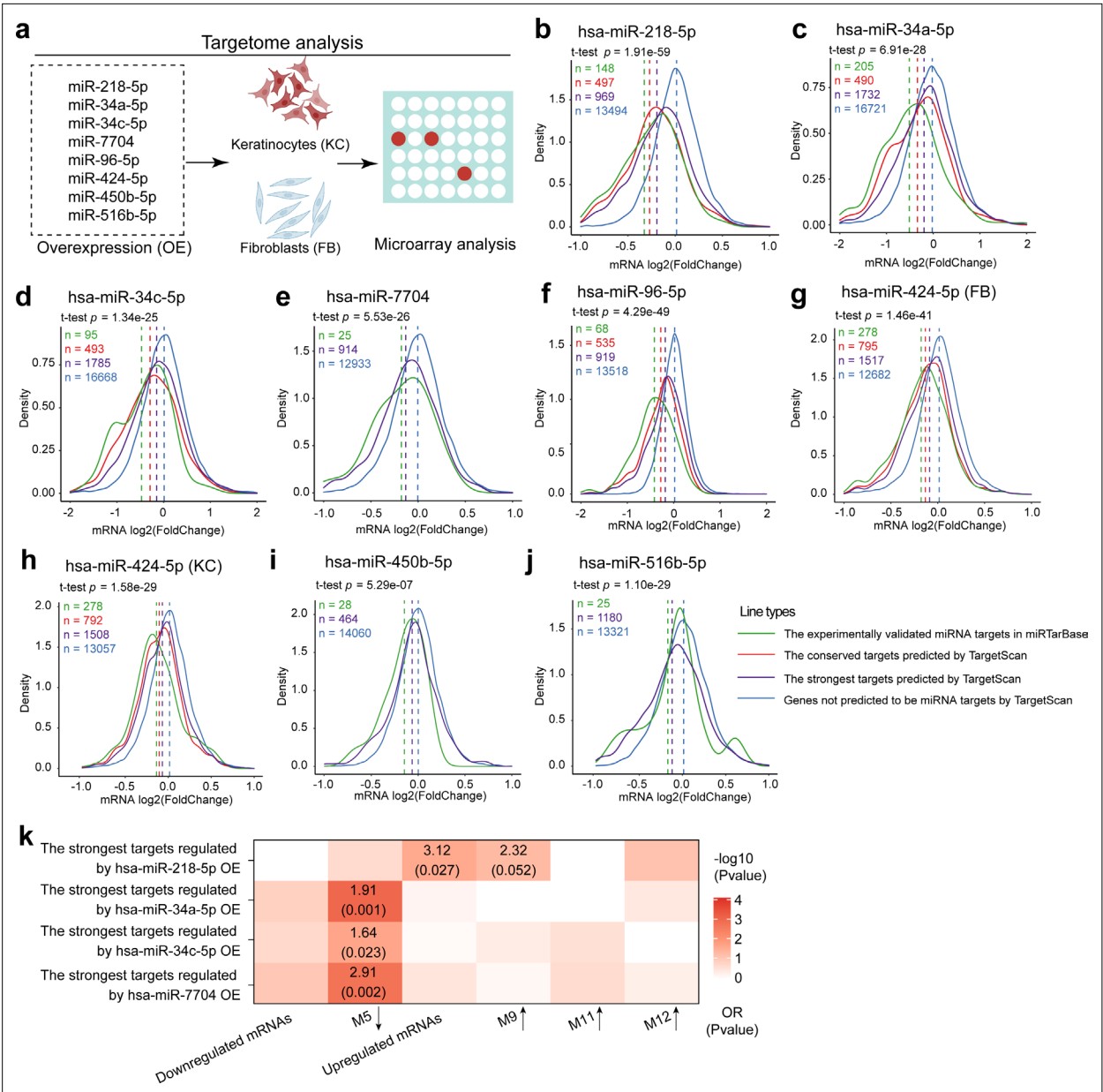

**Figure 6.** Targetome analysis of the dysregulated miRNAs in human primary keratinocytes or fibroblasts. (**a**) A schematic diagram of targetome analysis. Microarray analysis was performed in keratinocytes (KC) or fibroblasts (FB) with miR-218-5p (**b**), miR-34a-5p (**c**), miR-34c-5p (**d**), miR-7704 (**e**), miR-96-5p (**f**), miR-424-5p (**g, h**), miR-450-5p (**i**), or miR-516-5p (**j**) overexpression (OE). Density plots of mRNA log2(fold change) are shown. Wilcoxon *t*-tests were performed to compare the TargetScan predicted strongest targets (purple) with the nontargets (blue) for each of these miRNAs. The conserved and experimentally validated targets are marked with red and green colors, respectively. Dotted lines depict the average log2(fold change) values for each mRNA group. (**k**) A heatmap shows the enrichment for venous ulcer (VU)-affected mRNAs and mRNA modules in the targets of miR-218-5p, miR-34a-5p, miR-34c-5p, or miR-7704 validated by the microarray analysis. Odds ratio (OR) and p values are shown when OR >1 and p value <0.05 (Fisher's exact test).

The online version of this article includes the following figure supplement(s) for figure 6:

**Figure supplement 1.** Overexpression efficiency in primary keratinocytes or fibroblasts transfected with respective miRNAs. (a–g) qRT-PCR analysis of miRNAs in keratinocytes or fibroblasts transfected with miRNA mimics for 24 hr (*n* = 3–4).

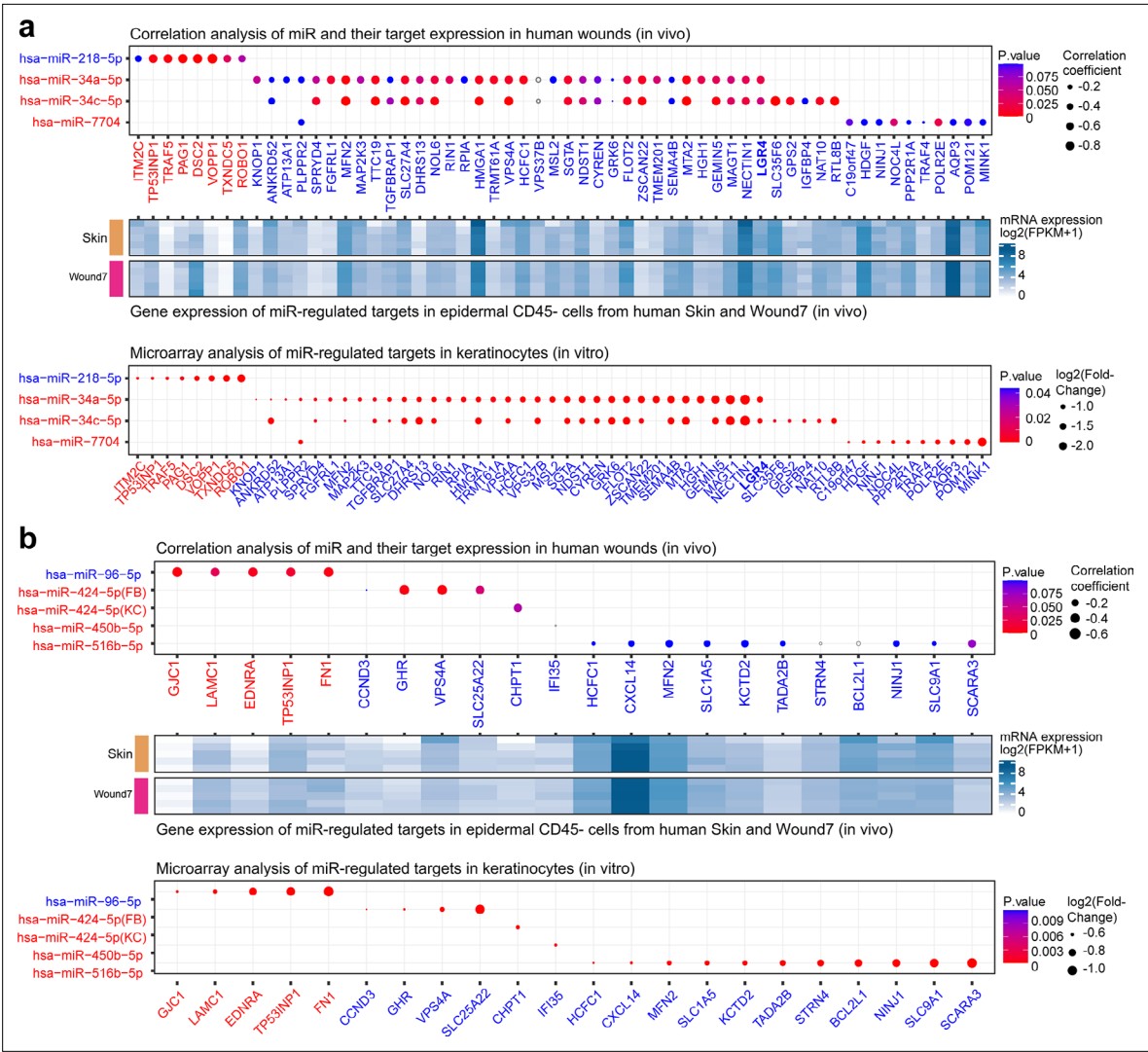

**Figure 7.** Analysis of the dysregulated miRNAs and their targets in vivo and in vitro. Pearson's correlation analysis of expression of the dysregulated miRNAs: (**a**) upper panels: miR-218-5p, miR-34a-5p, miR-34c-5p, and miR-7704; (**b**) upper panels: miR-96, miR-424, miR-450b, and miR-516b and their targets in RNA-seq data of human skin and wound samples. Gray circles indicate correlation coefficients >0. The gene expression of miRNA-regulated targets in epidermal CD45⁻ cells from human skin and wound day 7 (**a**, **b**: middle panels). The target mRNAs are significantly changed (fold change ≤−1.2 and p value <0.05) by the indicated miRNAs in the microarray analysis (**a**, **b**: lower panels).

mRNAs (Wilcoxon *t*-test p values: 1.34e−25–1.91e−59). The differences were more significant when we divided the strongest targets into conserved and experimentally validated subtypes, confirming the bioinformatics prediction robustness of the miR targets applied in this study (*Figure 6b–j*).

Notably, we observed significant enrichment (Fisher's exact test, OR >1, p value <0.05) of the experimentally validated miR-218-5p, miR-34a-5p, miR-34c-5p, and miR-7704 targets for the VU gene signature (*Figure 6k* and *Figure 5—source data 1*). We have previously shown that miR-34a and miR-34c inhibit keratinocyte proliferation and migration, while promoting apoptosis and inflammatory response, resulting in delayed wound repair in a mouse model (*Wu et al., 2020*). We validated robustness of the bioinformatics approach applied in this study by miR-34a/c reidentification. Here, we discovered that both the predicted (*Figure 4b*) and validated (*Figure 6k*) miR-34a/c targets were enriched for the downregulated M5 module mRNAs in VU. Notably, our microarray analysis in cultured keratinocytes overexpressing miR-34a or miR-34c confirmed that miR-34 reduced the expression of 39 hub genes in the M5 module (log2(fold change) ≤−0.58, p value <0.05, *Figure 7a*, lower panel), and 26 of them exhibited negative correlation (Pearson's r = −0.83 to −0.45, p value <0.05) with miR-34a/c expression levels in the human skin and wound samples (*Figure 7a*, upper panel and *Figure 5—source*

*data 1*), suggesting that they were miR-34a/c targets in vivo. MiR-218-5p was downregulated in the VU compared to the acute wounds and the skin (*Figure 5c* and *Figure 5—figure supplement 1*). Its predicted (*Figure 4b*) and validated (*Figure 7a*) targets were both enriched for the upregulated or M9 module mRNAs in VU. Among the 10 in vitro validated targets (*Figure 7a*, lower panel), eight negatively correlated (Pearson's *r* = −0.82 to −0.46, p value <0.05) with miR-218-5p expression in the human skin and wounds (*Figure 7a*, upper panel and *Figure 5—source data 1*). Interestingly, this study also identified miR-7704, a human-specific miR, with significantly increased expression in VU (*Figure 5e* and *Figure 5—figure supplement 1*). Similar to miR-34a/c, the predicted (*Figure 4b*) and validated (*Figure 6k*) miR-7704 targets were highly enriched for the M5 module mRNAs downregulated in VU.

For miR-96-5p, miR-424-5p, miR-450-5p, and miR-516b-5p, although their predicted targets were significantly enriched (Fisher's exact test: OR >1, p value <0.05) for VU-associated mRNAs (*Figure 4b*), we did not find similar enrichment for their experimentally validated targets. Nevertheless, these miRs regulated some VU-associated hub genes in vitro (*Figure 7b*, lower panel) and also exhibited an anticorrelated expression pattern with their targets in vivo (*Figure 7b*, upper panel), for example, miR-96-5p from the downregulated m9 module targets the M9 mRNAs upregulated in VU, including TP53INP1, LAMC1, EDNRA, GJC1, and FN1; while miR-424-5p from the upregulated m8 miR module targets the M5 mRNAs downregulated in VU, including SLC25A22, VPS4A, and GHR (*Figure 5—source data 1*).

In line with the in vivo expression of these eight miRs (i.e., miR-218-5p, miR-96-5p, miR-424-5p, miR-450-5p, miR-516b-5p, miR-34a-5p, miR-34c-5p, and miR-7704) in the epidermal cells of human skin and wounds (*Figure 5k*), we also confirmed the in vivo expression of their experimentally validated targets in the same cell type by RNA-seq (*Figure 7a, b* middle panels).

Together, our validation work focusing on epidermal keratinocytes confirmed the robustness and reproducibility of this dataset and highlighted its value as a reference for studying the physiological and pathological roles of miRs in human skin wound healing.

## Cooperativity of VU pathology-relevant miRs

From the miR-mediated gene expression regulation networks underpinning VU pathology (*Figure 4—figure supplements 1 and 2*, and *Figure 4—source data 2*), we caught a glimpse of presumable miR cooperativity through targeting the same mRNAs, that is, cotargeting among miRs, which reportedly imposing stronger and more complex repression patterns on target mRNA expression (*Cherone et al., 2019*). For the miRs with unrelated seed sequences, we found that miR-34a/c and miR-424-5p or miR-7704 shared 8–10 targets, and these miRs were coexpressed in the m8 module. We showed that among the downregulated miRs in VU, miR-96-5p and miR-218-5p shared eight targets.

In addition, we performed functional annotations for the genes regulated by the VU-associated miRs identified in the microarray analysis (*Figure 8a* and *Figure 8—source data 1*). Both miR-218-5p and miR-96-5p were predicted to promote ribosome biogenesis and nc RNA processing, while miR-218-5p might also suppress keratinization. miR-34a/c-5p were predicted to enhance innate immune response, while reducing mitosis. Similarly, GO analysis indicated that miR-424-5p and miR-516b-5p might increase the cellular defense response, while inhibiting cell proliferation. In addition, we showed that miR-450-5p upregulated genes related to the ncRNA metabolic process and mitochondrial respiratory chain complex assembly, whereas miR-7704 downregulated insulin, ERBB, and small GTPase-mediated signaling pathway-related genes. Of particular interest, combining the miR expression changes with their annotated functions, we found a regular pattern, that is, the miRs upregulated in VU (i.e., miR-34a-5p, miR-34c-5p, miR-424-5p, miR-450-5p, miR-7704, and miR-516-5p) were predicted to promote inflammation but inhibit proliferation; whereas the miRs downregulated in VU (i.e., miR-218-5p and miR-96-5p) might be required for cell growth and activation (*Figure 8b*). Therefore, these VU-dysregulated miRs might cooperatively result in impaired cell migration and proliferation accompanied with persistent inflammation, contributing to the stalled wound healing characterized with failed transition from the inflammatory phase to the proliferative phase (*Landén et al., 2016*).

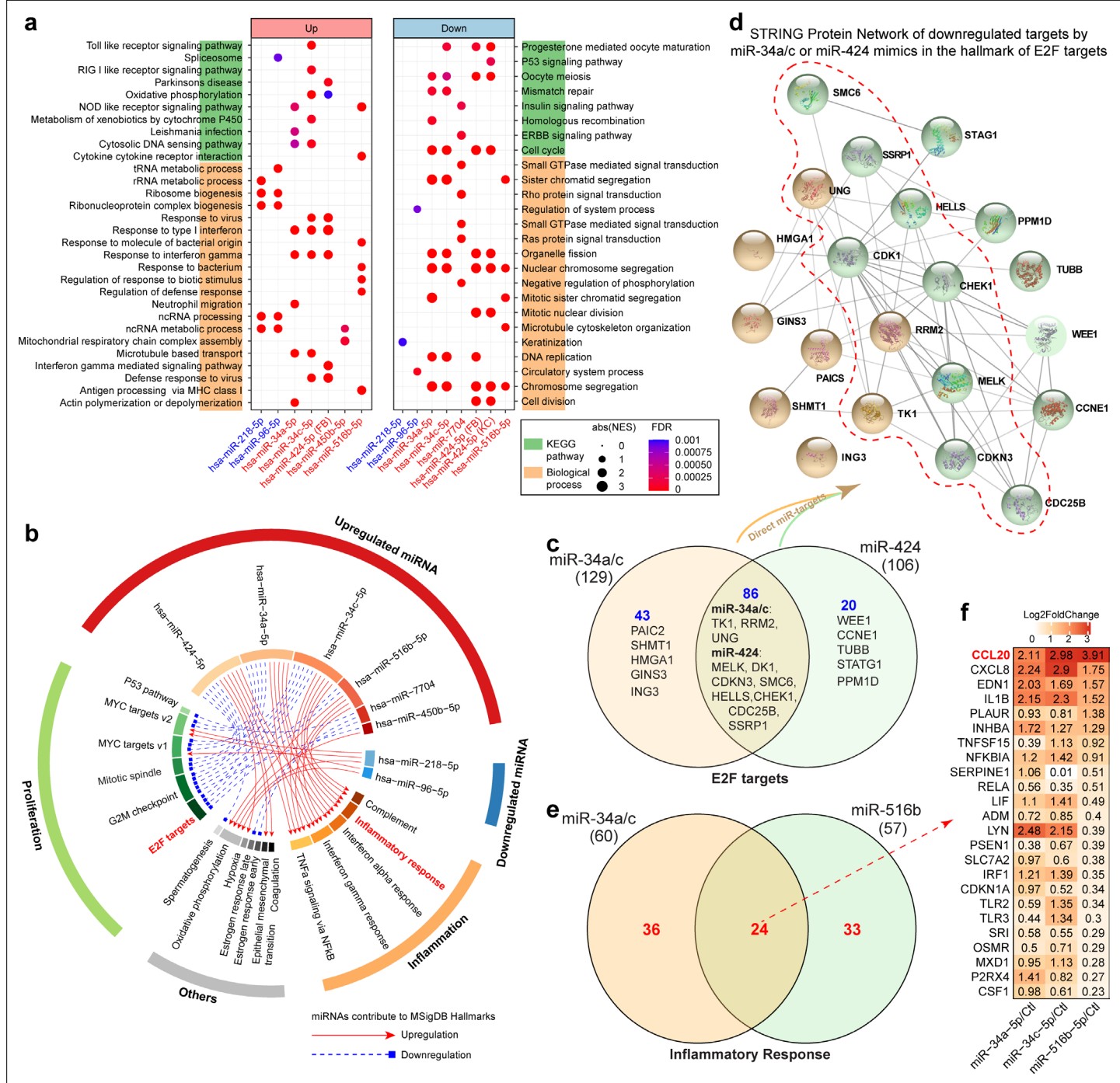

**Figure 8.** Cooperativity of venous ulcer (VU) pathology-relevant miRNAs. For the microarray data in keratinocytes (KC) or fibroblasts (FB) with miR-218-5p, miR-96-5p, miR-34a-5p, miR-34c-5p, miR-7704, miR-424-5p, and miR-516-5p overexpression, we analyzed Kyoto Encyclopedia of Genes and Genome (KEGG) pathways, biological processes (**a**), and Molecular Signatures Database (MSigDB) hallmarks (**b**) enriched by the genes up- or downregulated by these miRNAs. (**c**) Venn diagram shows the number of genes in the E2F target pathway regulated by miR-34a/c-5p and/or miR-424-5p. The direct targets of each miRNA are depicted as a STRING Protein Network in (**d**). The number of inflammatory response-related genes regulated by miR-34a/c-5p and miR-516b-5p and ranked log2fold changes of overlapped genes from microarray analysis are shown in (**e, f**).

The online version of this article includes the following source data for figure 8:

**Source data 1.** Gene ontology analysis of the miRNA-regulated genes in the microarray data.

# Cooperation of miR-34a, miR-424, and miR-516 in regulating keratinocyte migration, proliferation, and inflammatory response

To validate miR cooperativity in modulating the key pathological process in VU, we analyzed migration, proliferation, and inflammatory response of keratinocytes with individual miR or miR combination overexpression or inhibition. The microarray data GO analysis (*Figure 8b*) showed that three miRs upregulated in VU could suppress the expression of E2F target genes (*Figure 8c*): miR-34a/c-5p reduced the level of 129 mRNAs (including eight miR-34a/c targets, *Figure 8d*), while miR-424-5p downregulated the expression of 106 mRNAs (including 13 miR-424 targets, *Figure 8d*). Although 86 mRNAs were commonly regulated by miR-34a/c-5p and miR-424-5p, none of them were cotargeted by these miRs (*Figure 8c*). E2F signaling plays a unique role in keratinocyte proliferation and migration, as well as in wound repair and epidermal regeneration in vivo (*D'Souza et al., 2002*). Similarly, in the cell cycle pathway, miR-34a-5p directly targeted CCND1, CDK6, HDAC1, and E2F3, while miR-424-5p targeted ANAPC13, CCNE1, CDC25B, CDK1, CDKN1B, CHK1, WEE1, and YWHAH, and only CDC23 was cotargeted by both miRs (*Figure 9—figure supplement 1a*). We thus hypothesized that miR-34a-5p and miR-424-5p might cooperate to impact stronger on cell proliferation and migration by targeting different gene sets within the same signaling pathway. To test this idea, we measured keratinocyte growth by detecting proliferation marker gene Ki-67 expression both on mRNA and protein levels. We found that although miR-34a-5p or miR-424-5p alone could reduce Ki-67 levels, their combination suppressed stronger Ki-67 expression (*Figure 9a, b*, *Figure 9—figure supplement 1b*, and *Figure 9—source data 1*). The cooperativity between miR-34a-5p and miR-424-5p in repressing keratinocyte growth was further confirmed by comparing cell growth curves generated with a live cell imaging system (*Figure 9c*, *Figure 9—figure supplement 1c*, and *Figure 9—video 1*). Next, with scratch wound healing assays, we showed that simultaneous overexpression of miR-34a-5p and miR-424-5p delayed keratinocyte migration, whereas inhibition of endogenous miR-34a-5p or miR-424-5p enhanced keratinocyte motility (*Figure 9d–g*).

Moreover, our microarray analysis of keratinocytes with miR overexpression showed that the miR-34a-5p and miR-516b-5p combination extended the list of inflammation-related upregulated genes (*Figure 8e*). In particular, both miRs enhanced keratinocyte expression of inflammatory chemokines/cytokines, for example, CCL20, CXCL8, and IL1B (*Figure 8f*). We showed that simultaneously overexpressing miR-34a-5p and miR-516b-5p induced a higher CCL20 expression compared to the individual overexpression of each miR, suggesting their cooperativity in promoting inflammation (*Figure 9h*).

In summary, our study identified VU signature miRs, for example, miR-34a, miR-424, and miR-516, with cooperativity in inflicting more severe pathological changes (*Figure 9i*). These findings open new opportunities of developing wound treatment targeting cooperating miRs with potentially higher therapeutic efficacy and specificity.

## Discussion

Our genome-wide paired analysis of miR and mRNA expression in human healing and nonhealing wounds provides a novel global landscape of the miR regulatory roles in wound biology. A detailed overview of the mRNA expression context at different healing stages or under pathological condition VU allows a more precise understanding about the complex role of miRs in wound repair. The same miR is often described to play different or even opposite roles in different cells, as each cell type has specific gene expression context subjected to the miR-mediated posttranscriptional regulation (*Erhard et al., 2014*). Thereby, the different mRNA expression profiles in acute or chronic wounds should be considered to understand the precise role of an miR in these distinct contexts. With this aspect in mind, we highlight miRs with their targetome most enriched in the VU mRNA signature, as these miRs display a higher likelihood to regulate pathologically relevant genes. Notably, certain of these miRs did not exhibit the greatest expression change in VU, they would thus be missed with the commonly used strategy that focuses on the top miR expression profiling data changes.

Another strength of our study is the decryption of time-resolved miR–mRNA expression pattern during human skin wound healing, providing a temporal view to our understanding of the functional miR roles. miRs and their target gene expression contexts change dynamically to support different functional needs during wound repair. Defining an miR as 'pro-healing' or 'anti-healing' requires

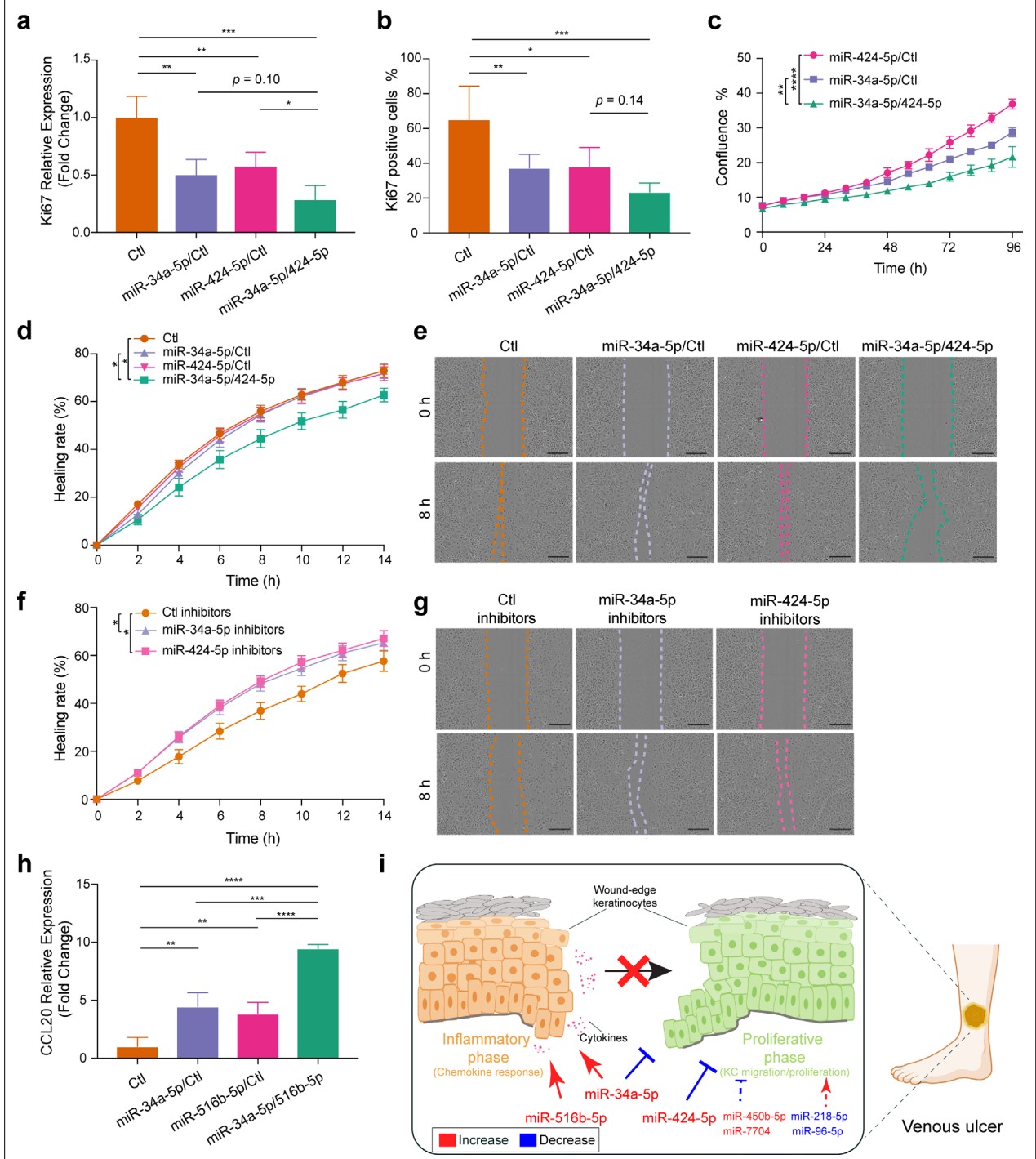

**Figure 9.** Cooperation of miR-34a, miR-424, and miR-516 in regulating keratinocyte proliferation, migration, and inflammatory response. Ki-67 expression was detected in keratinocytes transfected with miR-34a-5p or miR-424-5p mimics alone or both mimics for 24 hr ($n = 3$) by qRT-PCR (**a**) and immunofluorescence staining ($n = 4$–6) (**b**). (**c**) The growth of the transfected keratinocytes ($n = 3$) was analyzed with a live cell imaging system. The migration of keratinocytes transfected with miR-34a-5p or miR-424-5p mimics alone or both mimics (**d**, **e**) ($n = 8$–10) or miRNA inhibitors (**f**, **g**) ($n = 10$) was analyzed with a live cell imaging system (scale bar, 300μm). (**h**) qRT-PCR analysis of CCL20 in keratinocytes transfected with miR-34a-5p or miR-516b-5p mimics alone or both mimics for 24 hr ($n = 3$). (**i**) Proposed mechanism by which venous ulcer (VU)-dysregulated miRNAs cooperatively contribute to the stalled wound healing characterized with failed inflammation–proliferation transition (schematic generated by BioRender). *$p < 0.05$; **$p < 0.01$; ***$p < 0.001$; and ****$p < 0.0001$ by one-way analysis of variance (ANOVA) with uncorrected Fisher's least significant difference (LSD) (**a**, **b**, **h**) and two-way ANOVA (**c**, **d**, **f**). Data are presented as mean ± standard deviation (SD) (**a–c**, **h**) or mean ± standard error of the mean (SEM) (**d**, **f**).

*Figure 9 continued on next page*

*Figure 9 continued*

The online version of this article includes the following video, source data, and figure supplement(s) for figure 9:

**Source data 1.** Cooperation of miR-34a, miR-424, and miR-516 in regulating keratinocyte proliferation and inflammatory response.

**Figure supplement 1.** Cooperation of miR-34a and miR-424 in regulating keratinocyte proliferation.

**Figure 9—video 1.** Keratinocyte growth was analyzed with a live cell imaging system.

https://elifesciences.org/articles/80322/figures#fig9video1

specifying its temporal expression pattern. For example, continuous expression of a miR that is beneficial for one healing phase but not the other might also lead to deleterious effects.

To understand the molecular mechanisms underlying the miR coexpression patterns, we analyzed the enriched TFs for each miR module with experimentally validated TF-miR regulation data (*Tong et al., 2019*). This analysis led us to important TFs, such as GATA3 (*Kaufman et al., 2003*; *Kurek et al., 2007*) that play fundamental roles in skin development and postnatal remodeling, as well as KLF4, crucial for establishing skin barrier function (*Segre et al., 1999*). Notably, both GATA3 and KLF4 are reportedly downregulated in human VU (*Stojadinovic et al., 2014*; *Stojadinovic et al., 2008*). Our study confirms these findings and provides a novel insight, showing that the loss of these TFs might contribute to VU pathology through their regulated miRs.

We summarized the miRNAs that reportedly regulate wound healing and found that most of them also surfaced in our study, supporting the robustness of our profiling data and bioinformatics analysis (*Appendix 1—figure 1* and *Supplementary file 3*). Thereby, our data would be potentially helpful to evaluate the clinical relevance of these miRs. For example, miR-34a/c reportedly enhance keratinocyte inflammatory response, while suppressing proliferation and migration in cultured cells and mouse wound models (*Wu et al., 2020*). miR-34a was also identified as one of the most induced miRs in diabetic foot ulcer fibroblasts. Induction of miR-34a together with miR-21-5p and miR-145-5p inhibits fibroblast movement and proliferation, whereas activates cell differentiation and senescence (*Liang et al., 2016*). In this study, we described that miR-34a/c were specifically upregulated in VU, whereas their levels during wound repair remained relatively low and stable, suggesting their specific role in wound pathology. miR-34 targets were enriched in the M5 mRNA module, containing genes upregulated in the inflammatory phase of wound healing but downregulated in VUs. The VU-relevant miR-34 targetome identified in this study would be potentially useful for determining the precise role of miR-34 in VU pathology. Our current findings in human samples together with previous functional data (*Liang et al., 2016*; *Wu et al., 2020*) suggest that miR-34 inhibition along with modulation of additional deregulated miRs might be a promising VU treatment approach.

In addition, certain of these VU-related miRs are involved in skin-related functions but have not yet been linked to wound healing. For example, miR-218-5p regulates hair follicle development (*Zhao et al., 2019*), inhibits melanogenesis (*Guo et al., 2014b*), and enhances fibroblast differentiation (*Guo et al., 2014a*). miR-7704 was identified as an exosomal miR produced by melanocytes (*Shen et al., 2020*). miR-424-5p suppresses keratinocyte proliferation (*Ichihara et al., 2011*) and cutaneous angiogenesis (*Nakashima et al., 2010*; *Yang et al., 2017*). Moreover, our VU-related miR list (*Figure 4b*) also contains miRs without prior knowledge in their role either in skin or wound healing, for example, miR-517a-3p, miR-517b-3p, miR-516b-5p, miR-512-3p, and miR-450-5p. It would be highly interesting to examine the role of these miRs in VU. Overall, our dataset can serve as a valuable reference for prioritizing clinically relevant miRs for further functional studies.

Moreover, we studied the relationships between the dysregulated miRs in VU, regarding their target repertoire and biological functions and identified miRs that could act cooperatively. Such knowledge is required for developing combined miR therapeutics with increased specificity and efficacy (*Lai et al., 2019*). In the miR-target networks underpinning VU (*Figure 4—figure supplements 1 and 2*), we identified a few putative cooperating miR pairs/clusters that were coexpressed and shared multiple common targets, including the upregulated miR-34a/c together with miR-424-5p and miR-7704, as well as the downregulated miR-218-5p and miR-96-5p in VU. Furthermore, although not sharing targets, the majority of the VU-dysregulated miRs could still regulate the common BPs coordinately. For example, the miRs upregulated in VU (e.g., miR-34a/c-5p, miR-424-5p, miR-450-5p, miR-7704, and miR-516-5p) were predicted to promote inflammation but inhibit proliferation; whereas the miRs downregulated in VU (e.g., miR-218-5p and miR-96-5p) might be needed for cell growth

and activation. Particularly, we validated the cooperation between miR-34a-5p and miR-424-5p in suppressing keratinocyte proliferation and migration, as well as the cooperation between miR-34a-5p and miR-516b-5p in enhancing keratinocyte expression of inflammatory chemokine CCL20. The effect of miR-34 family in promoting keratinocyte inflammatory response has been recently reported and one of the miR-34 targets, LGR4, has been identified to mediate this effect (*Wu et al., 2020*). However, the mechanism underlying the proinflammatory function of miR-516b-5p in keratinocytes warrants further study. As a combined consequence, this VU-miR signature could disrupt the swift transition from inflammation to proliferation (*Figure 9i*). The failure of this phase transition represents a core pathology of chronic wounds (*Eming et al., 2014*; *Landén et al., 2016*). Our findings open the possibility of developing innovative wound treatment targeting multiple pathologically relevant cooperating miRs to attain higher therapeutic efficacy and specificity.

Based on the integrative small and long RNA-omics analysis of human wound tissues, we have developed an openly available compendium (https://www.xulandenlab.com/humanwounds-mirna-mrna) for the research community. This novel, rich resource enabled us to gain a network view of miR-mediated gene regulation during human physiological and pathological wound repair in vivo. With the same sequencing datasets, we have also analyzed circular RNA expression and their potential interaction with miRs and miR targets (*Toma et al., 2022*), which results can be queried at https://www.xulandenlab.com/humanwounds-circrna. These efforts result in many testable hypotheses for future studies elucidating gene expression regulatory mechanisms underpinning tissue repair.

A limitation of bulk RNA-seq of wound tissues is that the observed expression changes could occur in a single or multiple cell types or reflect cell composition changes. To obtain cell-specific miRNA expression data, single-cell sRNA-seq is required. However, this technology needs extensive cell handing and therefore has only been applied to few cells and still remains challenging to be used at a scale for analyzing complex dynamics of tissue (*Nielsen and Pedersen, 2021*). Alternatively, we analyzed miR and mRNA expression in epidermal cells isolated from human skin and wound tissues and validated miR-mediated gene regulation in keratinocytes, excluding the possibility of cellular heterogeneity-related changes for several miRs identified in our study.

In conclusion, this genome-wide, integrative analysis of miR and mRNA expression in human skin and wound tissues reinforce and extend the evidence about the functional role of miRs in wound repair and their therapeutic potential for chronic wound treatment. By combining miR expression patterns with their specific target gene expression context, we identified miRs highly relevant to VU pathology. This rigorous and in-depth molecular characterization of human wound tissues adds a novel dimension to our current knowledge mostly relying on nonhuman models and would serve as a unique platform and valuable resource for further mechanistic studies of miRs with a high translational potential.

## Materials and methods

### Human wound samples collection

Patients with VUs, which persisted for more than 4 months despite conventional therapy, were enrolled in this study (*Table 2*). Tissue samples were collected from the lower extremity at the nonhealing edges of the ulcers by using a 4-mm biopsy punch (*Figure 1a*). Each VU biopsy contains about 50% wound edge with epidermis and 50% wound-bed area. Healthy donors above 60 years old without skin diseases, diabetes, unstable heart disease, infections, bleeding disorder, immune suppression, and any ongoing medical treatments were recruited (*Table 3*). Two full-thickness excisional wounds (4 mm in diameter) were created at the lower extremity on each donor, and the excised skin was saved as intact skin control (Skin). With a 6-mm biopsy punch, we excised the entire wounds (including the ring-shape wound edges covered with epidermis and the center wound beds with granulation tissues) at day 1 (Wound1) and day 7 (Wound7) after wounding (*Figure 1a*).

**Table 3.** Characteristics of the healthy donors.

| Donor | Sex | Age (years) | Ethnicity | Wound location | Experiment |
|---|---|---|---|---|---|
| 1 | F | 66 | Caucasian | Lower leg | RNA-seq |
| 2 | M | 69 | Caucasian | Lower leg | RNA-seq |
| 3 | F | 67 | Caucasian | Lower leg | RNA-seq |
| 4 | M | 69 | Caucasian | Lower leg | RNA-seq and qRT-PCR |
| 5 | F | 64 | Caucasian | Lower leg | RNA-seq and qRT-PCR |
| 6 | F | 60 | Caucasian | Lower leg | qRT-PCR |
| 7 | F | 66 | Caucasian | Lower leg | qRT-PCR |
| 8 | F | 60 | Caucasian | Lower leg | qRT-PCR |
| 9 | F | 67 | Caucasian | Lower leg | qRT-PCR |
| 10 | F | 65 | Caucasian | Lower leg | qRT-PCR |
| 11 | F | 66 | Caucasian | Lower back | Cell isolation; RNA-seq |
| 12 | M | 69 | Caucasian | Lower back | Cell isolation; RNA-seq |
| 13 | F | 67 | Caucasian | Lower back | Cell isolation; RNA-seq |
| 14 | M | 69 | Caucasian | Lower back | Cell isolation; RNA-seq |
| 15 | F | 64 | Caucasian | Lower back | Cell isolation; RNA-seq |
| 16 | F | 42 | Caucasian | Lower back | LCM |
| 17 | M | 58 | Caucasian | Lower back | LCM |
| 18 | M | 42 | Caucasian | Lower back | LCM |
| 19 | M | 28 | Caucasian | Lower back | LCM |
| 20 | M | 30 | Caucasian | Lower back | LCM |
| 21 | F | 46 | Caucasian | Lower back | LCM |
| 22 | F | 47 | Caucasian | Lower back | LCM |

M, male; F, female.

## Sample preparation, RNA extraction, library preparation, and sequencing

### Laser capture microdissection

After embedding of the snap frozen skin and wound biopsies, 8-µm tissue sections were stained with hematoxylin. LCM was then performed with Leica LMD7000 (Leica, Bernried, Germany) to separate the epidermis from each section.

### Magnetic Activation Cell Sorting for CD45⁻ epidermal cell

The fresh skin and wound tissues were washed in PBS and incubated in dispase II (5 U/ml, Thermo Fisher Scientific) at 4°C overnight, and the epidermis was separated from the dermis as previously described (*Henrot et al., 2020*). After the digestion in 0.025% trypsin/ethylenediaminetetraacetic acid (EDTA) solution at 37°C for 15 min, CD45⁻ and CD45⁺ epidermal cells were separated by using CD45 microbeads and MACS MS magnetic columns according to the manufacturer's instructions (Miltenyi Biotec, Bergisch Gladbach, Germany).

### RNA extraction

Snap frozen tissue samples were homogenized with the TissueLyser LT (Qiagen), and total RNA was isolated using the miRNeasy Mini kit (Qiagen). RNA quality and quantity were determined by using

Agilent 2100 Bioanalyzer (Agilent Technologies) and Nanodrop 1000 (Thermo Fisher Scientific Inc), respectively.

## Small RNA library preparation and sequencing

The sRNA-seq libraries were constructed using 3 µg total RNA per sample and NEB Next Multiplex Small RNA Library Prep Set for Illumina (NEB) following the manufacturer's recommendation. Briefly, total RNA was first ligated to adaptors at the 3′ end by NEB 3′ SR adaptor and 5′ end by T4 RNA ligase followed by reverse transcription into cDNA using M-MuLV Reverse Transcriptase. PCR amplification of cDNA was performed using SR primers for Illumina and index primers. The PCR products were purified, and DNA fragments spanning from 140 to 160 bp were recovered and quantified by DNA High Sensitivity Chips on the Agilent Bioanalyzer. The libraries were sequenced on an Illumina Hiseq 2500 platform (Illumina, Inc) using single-end 50 bp reads, and all samples were run side by side.

## mRNA library preparation and sequencing

The long RNA-seq libraries were constructed with a total amount of 2 µg RNA per sample. First, the ribosomal RNA was depleted by Epicentre Ribo-zero rRNA Removal Kit (Epicentre). Second, strand-specific total-transcriptome RNA-seq libraries were prepared by incorporating dUTPs in the second-strand synthesis step with NEB Next Ultra Directional RNA Library Prep Kit for Illumina (NEB). The CD45⁻ epidermal keratinocytes RNA-seq libraries were constructed by following the tutorial of NuGen Ovation Solo RNA-Seq System (Human part no. 0500). Finally, the libraries were sequenced on an Illumina Hiseq 4000 platform, and 150-bp paired-end reads were generated for the following analysis.

## Analysis of miRNA-sequencing data

### Quality control, mapping, and quantification

Quality of raw data was assessed using FastQC v0.11.8 (http://www.bioinformatics.babraham.ac.uk/projects/fastqc/). We used mapper.pl module in the miRDeep2 v0.1.3 package (*Friedländer et al., 2012*, *Mackowiak, 2011*) to filter low-quality reads and remove sequencing adaptors and redundancies. Trimmed reads with lengths greater than 18 nucleotides were mapped to GENCODE human reference genome (hg38) by the software Bowtie v1.2.2 (*Langmead et al., 2009*). The miRDeep2.pl module was then performed with default parameters to identify known miRNAs, which were compared to miRNAs in miRBase release 22.1 (*Kozomara et al., 2019*). Counts of reads mapped to each known mature miRNAs were acquired from the quantifier.pl module output without allowing mismatch. miRNAs with read counts less than five in more than half of twenty samples were discarded since these miRNAs are unlikely to give stable and robust results. Raw counts of 562 miRNAs were normalized for sequencing depth using TPM methods (transcript per million = mapped read count/total reads * 10e6) (*Zhou et al., 2010*) and prepared for further analysis.

### DE analysis

The DESeq2 workflow (*Love et al., 2014*) was carried out to fit raw counts to the negative binomial generalized linear model and to calculate the statistical significance of each miRNA in each comparison. In particular, the paired model was employed when comparing samples from the same donor. p *values* obtained from the Wald test were corrected using Benjamini–Hochberg (BH) multiple test to estimate the FDR. The DE miRNAs were defined as FDR <0.05 and |log2(fold change)| ≥1.

### Principal component analysis

To explore the similarity of each sample, principal component analysis (PCA) was performed by using a DESeq2 built-in function *plotPCA* on the transformed data, in which the variances and size factors were stabilized and corrected. PCA and heatmaps were plotted by using *ggplot2* (*Hadley, 2016*) and *ComplexHeatmap* (*Gu et al., 2016*) packages in RStudio (https://rstudio.com/).

### Weighted gene coexpression network analysis

The normalized expression of 562 miRNAs was used as input to the WGCNA R package (*Langfelder and Horvath, 2008*). First, we calculated the strength of pairwise correlations between miRNAs using

the 'biweight' mid-correlation method. The function *pickSoftThreshold* was then employed to compute the optimized soft-thresholding power based on connectivity, which led to an approximately scale-free network topology (*Zhang and Horvath, 2005*). Second, a signed weighted coexpression network was constructed with a power of 18 using the one-step *blockwiseModules* algorithm (*Figure 2—figure supplement 2a*). Network modules were filtered according to parameters: minModuleSize = 10 and mergeCutHeight = 0.25.

The expression profile of each module was represented by the module eigengene (ME), referred to as the PC1 of all miRNAs in each module. Pearson's correlations (values from −1 to 1) and the corresponding p values between MEs and traits were computed. p values were further adjusted to FDR across all the modules using the BH method. Modules significantly associated with each trait were selected by FDR <0.05 and absolute correlation coefficients >0.4. The module membership (also known as kME) of each miRNA was calculated by the correlations between miRNA expression and ME. MiRNAs with the highest kME values were defined as intramodular hub miRNAs, and networks of hub miRNAs in significant modules were visualized using the Cytoscape v3.7.2 software (*Shannon et al., 2003*).

To check the robustness of module definition, we carried out module preservation analysis and calculated the standardized *Z*-scores for each module by permutating 200 times using the same 20 samples as reference and test datasets. Modular preservation is strong if *Z*-summary >10, weak to moderate if 2 < *Z*-summary < 10, no evidence of preservation if *Z*-summary ≤2 (*Langfelder et al., 2011*).

## TF enrichment analysis

We leveraged a curated database about TF-miRNA regulations, TransmiR v2.0 (*Tong et al., 2019*), to identify the TFs regulating miRNA expression in each module. Fisher's exact tests were employed to evaluate the enrichment of each TF in the significant modules, and FDRs were adjusted to the total number of TFs (OR >1 and FDR <0.05). Correlations of gene expression between TFs and miRNA modules (represented by MEs) were further filtered to identify putative TF-mediated miRNA gene expression patterns (Pearson's correlation: p value <0.05, coefficient >0). The 562 miRNAs abundantly expressed in our samples were treated as the background dataset.

## Analysis of mRNA-sequencing data

Raw reads of mRNA-sequencing were first trimmed for adaptors and low-quality bases using Trimmomatic v0.36 software (*Bolger et al., 2014*). Clean reads were aligned to the human reference genome (GRCh38.p12), coupled with the comprehensive gene annotation file (GENCODEv31) using STAR v2.7.1a (*Dobin et al., 2013*). Gene expression was then quantified by counting unique mapped fragments to exons by using the feature count function from the Subread package (*Liao et al., 2014*). Raw counts for each gene were normalized to fragments per kilobase of a transcript, per million mapped reads (FPKM)-like values. Only mRNAs with FPKM ≥1 in at least 10 samples were kept for the rest analysis. We used the same pipeline described above for mRNA DE and PCA analysis. The DE mRNAs were defined as FDR <0.05 and |log2(fold change)| ≥0.58. WGCNA was carried out for 12,069 mRNAs with the optimal threshold power of 12 according to a fit to the scale-free topology of the coexpression network (*Figure 2—figure supplement 4a*). Thirteen mRNA modules were identified with the settings: maxBlockSize = 20,000, minModuleSize = 100, and mergeCutHeight = 0.25. Furthermore, mRNA module-enriched TF analysis was performed with a manually curated TF-target regulatory relationship database, TRRUST v2 (*Han et al., 2018*), using Fisher's exact tests. TFs with FDR <0.05 and OR >1 and the expression significantly correlated with respective mRNA modules (Pearson's correlation p value <0.05) were identified.

## GO analysis

We carried out GO analysis for mRNAs by using the WebGestalt tool (http://www.webgestalt.org/) (*Liao et al., 2019*), which applied a hypergeometric test in target and reference gene sets. GO terms of nonredundant BP with gene number less than 10 and adjust p value (FDR) >0.05 were filtered out.

## Integrative analysis of miRNA and mRNA expression changes in wound healing

### Expression correlation between mRNA and miRNA modules

An integrative analysis was carried out by relating the PC1 of miRNA expression, calculated using the *moduleEigengenes* function of WGCNA package (*Langfelder and Horvath, 2008*), to the PC1 of mRNA expression in each module. The miRNA-mRNA module pairs with a Pearson's correlation coefficient $<-0.5$ and a p value $<0.05$ were selected for the following enrichment analysis.

### Prediction of miRNA targets

We predicted both conserved and nonconserved target sites for all the 562 miRNAs by using the *get_multimir* function from R package multiMiR (*Ru et al., 2014*) (http://multimir.org/) based on the latest TargetScan v7.2 database (*Agarwal et al., 2015*, *Lewis et al., 2005*). All predicted miRNA targets were sorted by a primary score calculated for target site strength, and the top 25% with summed context++score $\leq-0.15$ were defined as the strongest miRNA targets. Targets that were not detected by the long RNA-seq were removed.

### Gene set enrichment analysis of miRNA targets in mRNA modules

We evaluated the degree of enrichment of miRNA modules' targets in mRNA modules. For this, we focused on the VU-specific DE miRNA, that is, the 22 up- and 10 downregulated miRNAs in VU compared to both the skin and acute wounds (FDR $<0.05$ and |log2(fold change)|$\geq$ 1), as well as the VU-associated modules' hub miRNAs, which kME values were greater than the median kME in respective modules (i.e., 14 miRNAs in m8, 9 miRNAs in m12, 20 miRNAs in m7, 29 miRNAs in m3, and 13 miRNAs in m9). Among these miRNAs' strongest targets, we selected the ones that were hit by $\geq$2 miRNAs from m8, m9, m12 modules or miRNAs downregulated in VU; $\geq$3 miRNAs from m7 module or miRNAs upregulated in VU; $\geq$4 miRNAs from m3 module, to capture putative module-driving targets. We performed gene set enrichment analyses for these miRs' targets in VU-specific DE mRNAs (FDR $<0.05$ and fold change $\geq$1.5) and VU-associated mRNA modules by using the R function *fisher.test*() based on the two-side Fisher's exact test (*Wu et al., 2016*). Furthermore, we performed enrichment analysis to identify individual miRNA with their strongest targets significantly enriched in VU-specific DE mRNAs or VU-associated mRNA modules (Fisher's exact test: OR $>1$, p value $<0.05$).

## Experimental validation of miRs' expression, targetome, and functions

### Quantitative RT-PCR

To detect miRNA, RNA from human skin and wounds was reverse transcribed using TaqMan Advanced miRNA cDNA Synthesis Kit (Thermo Fisher Scientific). Individual miRNA expression was then quantified using TaqMan Advanced miRNA Assays (Thermo Fisher Scientific) and normalized with miR-361-5p and miR-423-5p due to their relatively constant expression between human skin and wounds. To detect mRNA, we performed reverse transcription using the RevertAid First Strand cDNA Synthesis Kit (Thermo Fisher Scientific). Gene expression was examined by SYBR Green expression assays (Thermo Fisher Scientific) and normalized with housekeeping gene B2M and GAPDH. The primer sequences for B2M are forward primer (5'-AAGTGGGATCGAGACATGTAAG-3') and reverse primer (5'-GGAGACAGCACTCAAAGTAGAA-3'); GAPDH forward primer (5'-GGTGTGAACCATGAGAAGTA TGA-3') and reverse primer (5'-GAGTCCTTCCACGATACCAAAG-3').

### Primary cell culture and transfection

Adult human epidermal keratinocytes (Thermo Fisher Scientific) were cultured in EpiLife serum-free keratinocyte growth medium supplemented with Human Keratinocyte Growth Supplement (HKGS) and 100 units/ml penicillin and 100 μg/ml streptomycin (Thermo Fisher Scientific). Adult human dermal fibroblasts (Thermo Fisher Scientific) were cultured in Medium 106 supplemented with Low Serum Growth Supplement (LSGS) and 100 units/ml penicillin and 100 μg/ml streptomycin (Thermo Fisher Scientific). Cells were incubated at 37°C in 5% $CO_2$, and media was routinely changed every 2–3 days. Third passage keratinocytes at 50–60% confluence were transfected with 20 nM miRNA mimics (Horizon) or negative control using Lipofectamine 3000 (Thermo Fisher Scientific).

## Microarray analysis

Transcriptome profiling of keratinocytes and fibroblasts transfected with 20 nM miRNA mimics or control mimics for 24 hr (in triplicates) was performed using Affymetrix Genechip system at the Microarray Core facility of Karolinska Institute. Normalized expression data (log2 transformed value) were exported from Transcriptome Analysis Console (TAC) software and analyzed by the *limma* R package (*Ritchie et al., 2015*). In brief, expression data were first fitted to a linear model for each probe. Then, the empirical Bayes method was applied to compute the estimated coefficients of gene-wise variability and standard errors for comparisons of experimental and control groups. Genes with FC >1.2 and p value <0.05 between the miRNA mimics- and the control mimics-transfected cells were considered to be significantly changed. Gene set enrichment analysis, including BP, Kyoto Encyclopedia of Genes and Genomes (KEGG) pathway, and hallmark from Molecular Signatures Database (MSigDB) (http://www.gsea-msigdb.org/) (*Subramanian et al., 2005*), was performed with a ranked fold change list of all the genes by using the *fgsea* R package (*Korotkevich et al., 2021*) and visualized by using the *ggplot2* and *circlize* packages. Protein–protein interaction was analyzed using the STRING web resource (https://string-db.org/) (*Szklarczyk et al., 2019*).

## Immunofluorescence staining

Cells transfected with a combination of miRNA and/or control mimics (50 nM in total) were fixed in 4% paraformaldehyde for 15 min. Cells were incubated with the Ki-67 antibody (Cell Signaling Technology, RRID:AB_2687446) overnight at 4°C. The next day, cells were incubated with the secondary antibody conjugated with Alexa 488 (RRID:AB_2535792) for 40 min at room temperature. Cells were mounted with the ProLong Diamond Antifade Mountant with 4',6-diamidino-2-phenylindole (Thermo Fisher Scientific). Ki-67 signals were visualized with Nikon microscopy, and positive cells were counted using ImageJ software (National Institutes of Health).

## Cell proliferation assay

Cells were seeded in 12-well plates with a density of 20,000 cells/well. The plates were placed in IncuCyte live-cell imaging and analysis platform (Essen Bioscience) after cells attaching to the plates. Plates were imaged every 2 hr, and pictures were processed and analyzed using IncuCyte ZOOM 2018A software (Essen Bioscience).

## Migration assay

Human primary keratinocytes transfected with miRNA mimics or inhibitors were plated in Essen ImageLock 96-well plates (Essen Bioscience) at 15,000 cells per well for migration assay. Confluent cell layers were scratched using Essen wound maker to generate wounds. The cells were cultured with EpiLife medium without HKGS. The photographs were analyzed by using the IncuCyte software (Essen Bioscience).

## Statistical analysis

Sample size of each experiment is indicated in the figure legend. Data analysis was performed by using R and GraphPad Prism 7 software. Comparison between two groups was performed with Mann–Whitney *U*-tests (unpaired, nonparametric), Wilcoxon signed-rank test (paired, nonparametric), or two-tailed Student's *t*-test (parametric). Comparison between more than two groups was performed by one-way analysis of variance (ANOVA with uncorrected Fisher's LSD). The cell growth and migration assay were analyzed by using two-way ANOVA. p value <0.05 is considered to be statistically significant.

## Data availability

Raw data of sRNA-seq, long RNA-seq, and microarray performed in this study have been deposited to NCBI's Gene Expression Omnibus (GEO) database under the accession number GSE174661 and GSE196773, respectively. In addition, the analyzed dataset is presented with an online R Shiny app and can be accessed through a browsable web portal (https://www.xulandenlab.com/humanwounds-mirna-mrna). The analysis source code is available at https://github.com/Zhuang-Bio/miRNAprofiling; copy archived at swh:1:rev:14eaa943d868157d1de14b2e4ffc7f1be2552e15, *Liu, 2021*.

## Acknowledgements

We express our gratitude to the patients and healthy donors participating in this study. We thank Mona Ståhle, Desiree Wiegleb Edström, Peter Berg, Fredrik Correa, Martin Gumulka, and Mahsa Tayefi for clinical sample collection; Helena Griehsel for technical support. We thank the Microarray core facility at Novum, BEA, which is supported by the board of research at KI and the research committee at the Karolinska hospital. The computations/data handling was enabled by resources in the projects sens2020010 and 2021/22-701 provided by the Swedish National Infrastructure for Computing (SNIC) at UPPMAX, partially funded by the Swedish Research Council through grant agreement no. 2018-05973. Funding: This work was supported by Swedish Research Council (Vetenskapsradet, 2016-02051 and 2020-01400, Ning Xu Landén), Ragnar Söderbergs Foundation (M31/15, Ning Xu Landén), Welander and Finsens Foundation (Hudfonden, Ning Xu Landén), LEO foundation (Ning Xu Landén), Ming Wai Lau Centre for Reparative Medicine, Karolinska Institutet (Ning Xu Landén), and R01AR073614 from NIH/NIAMS (Marjana Tomic-Canic).

## Additional information

### Competing interests

Irena Pastar: is on the Board of Directors for the Wound Healing Society. Irena Pastar received payment for speaking at the Symposium of Advanced Wound Care and Wound Healing Society 2022 meeting, and the World Union of the Wound Healing Societies 2022. The author has no other competing interests to declare. Marjana Tomic-Canic: received grants from NIH/NIDDK [U01DK119085], NIH/NINR [R01NR015649], NIH/NIDDK [U24DK115255] and NIH/NIDDK [R41DK127900] for research unrelated to the topic of the manuscript. Marjana Tomic-Canic received payment for attending, speaking and moderating at the Symposium of Advanced Wound Care and Wound Healing Society 2022 meeting, and for attending the World Union of the Wound Healing Societies 2022. Marjana Tomic-Canic participates on the Scientific Advisory Board of Molnlycke. The patent No. 63/335,306 "Theraputic Compositions" is pending. The author has no other competing interests to declare. The other authors declare that no competing interests exist.

### Funding

| Funder | Grant reference number | Author |
|---|---|---|
| Swedish Research Council | 2016-02051 | Ning Xu Landén |
| Swedish Research Council | 2020-01400 | Ning Xu Landén |
| Ragnar Söderbergs Foundation | M31/15 | Ning Xu Landén |
| Welander and Finsens Foundation | | Ning Xu Landén |
| LEO foundation | | Ning Xu Landén |
| Ming Wai Lau Centre for Reparative Medicine | | Ning Xu Landén |
| Karolinska Institutet | | Ning Xu Landén |
| National Institutes of Health | R01AR073614 | Marjana Tomic-Canic |

No external funding was received for this work.

### Author contributions

Zhuang Liu, Letian Zhang, Conceptualization, Resources, Data curation, Software, Formal analysis, Validation, Investigation, Visualization, Methodology, Writing – original draft, Writing – review and editing; Maria A Toma, Dongqing Li, Xiaowei Bian, Resources, Validation, Investigation, Methodology; Irena Pastar, Marjana Tomic-Canic, Pehr Sommar, Resources, Supervision, Writing – review and editing;

Ning Xu Landén, Conceptualization, Resources, Supervision, Funding acquisition, Writing – original draft, Project administration, Writing – review and editing

### Author ORCIDs
Zhuang Liu http://orcid.org/0000-0001-8938-0086
Letian Zhang http://orcid.org/0000-0002-0987-0905
Ning Xu Landén http://orcid.org/0000-0003-4868-3798

### Ethics
Written informed consent was obtained from all the donors to collect and use the tissue samples. The study was approved by the Stockholm Regional Ethics Committee and conducted according to the Declaration of Helsinki's principles.

### Decision letter and Author response
Decision letter https://doi.org/10.7554/eLife.80322.sa1
Author response https://doi.org/10.7554/eLife.80322.sa2

---

## Additional files

### Supplementary files
- Supplementary file 1. Quality control of small RNA-sequencing data.
- Supplementary file 2. Quality control of rRNA-depleted total RNA-sequencing data.
- Supplementary file 3. Summary of microRNAs reported to regulate skin wound healing.
- Transparent reporting form

### Data availability
Sequencing data have been deposited in GEO under accession codes GSE174661 and GSE196773. The analyzed dataset is presented with an online R Shiny app and can be accessed through a browsable web portal (https://www.xulandenlab.com/humanwounds-mirna-mrna). The analysis source code is available at https://github.com/Zhuang-Bio/miRNAprofiling, copy archived at swh:1:rev:14eaa943d868157d1de14b2e4ffc7f1be2552e15. Source data files have been provided by excel files for figures 1c, 1d, 1e, 2a, 2b, 2c, 2d, 2e, 4a, 4b, 5b-j, 6k, 8, 9 and figure supplements 2-2, 2-4c, 6, 7a, b lower panels.

The following datasets were generated:

| Author(s) | Year | Dataset title | Dataset URL | Database and Identifier |
|---|---|---|---|---|
| Landén NX, Sommar P, Liu Z, Zhang L | 2022 | Integrative small and long RNA-omics analysis of human healing and non-healing wounds discovers cooperating microRNAs as therapeutic targets | https://www.ncbi.nlm.nih.gov/geo/query/acc.cgi?acc=GSE174661 | NCBI Gene Expression Omnibus, GSE174661 |
| Landén NX, Liu Z, Zhang L | 2022 | Integrative small and long RNA-omics analysis of human healing and non-healing wounds discovers cooperating microRNAs as therapeutic targets | https://www.ncbi.nlm.nih.gov/geo/query/acc.cgi?acc=GSE196773 | NCBI Gene Expression Omnibus, GSE196773 |

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

## Appendix 1

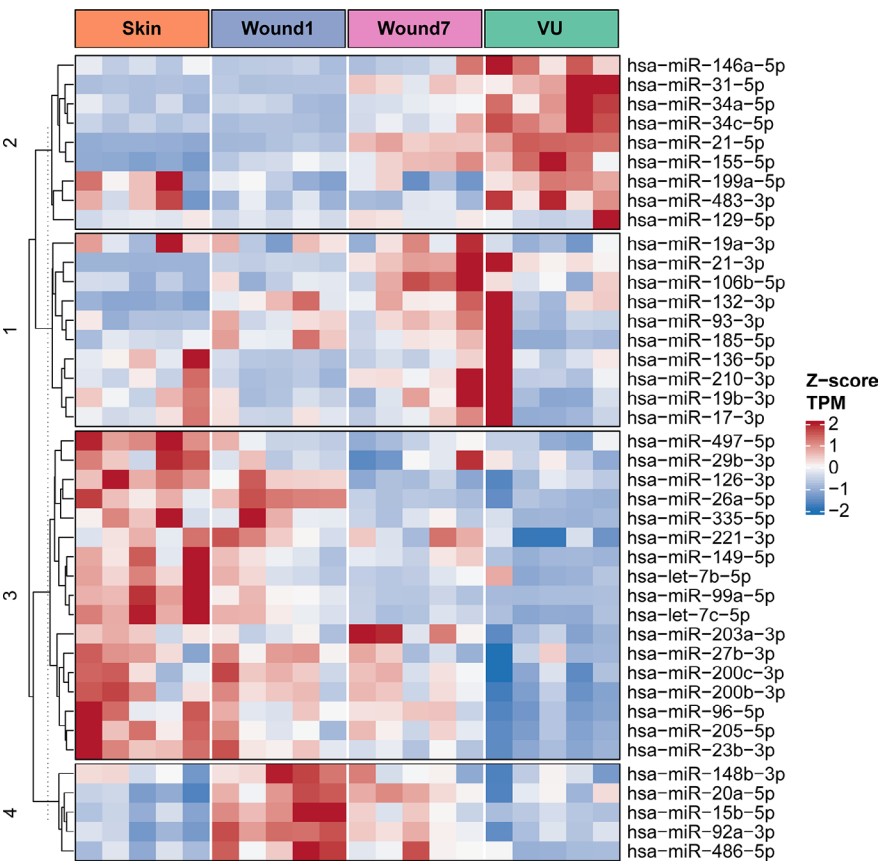

**Appendix 1—figure 1.** Summary of microRNAs reported to regulate skin wound healing.

