## [Editor Report]

A well-performed study looking at the comprehensive coding and non-coding RNA landscape of the healing wound in a highly controlled fashion. This study lends new insight into specific microRNAs as potential targets in human wound healing.

---

## [Decision Letter]

**Decision letter after peer review:**

[Editors’ note: the authors submitted for reconsideration following the decision after peer review. What follows is the decision letter after the first round of review.]

Thank you for submitting the paper "Integrative small and long RNA-omics analysis of human healing and non-healing wounds discovers cooperating microRNAs as therapeutic targets" for consideration by *eLife*. Your article has been reviewed by 3 peer reviewers, one of whom is a member of our Board of Reviewing Editors, and the evaluation has been overseen by a Senior Editor.

Comments to the Authors:

We are sorry to say that, after consultation with the reviewers, we have decided that this work will not be considered further for publication by *eLife*.

This study has conducted a comprehensive analysis to make the bioinformatic analysis results important to this field, while data volume regarding functional validation seems limited. The examined parameters are rather weak to reflect the regulatory network of the pathogenesis-relevant miRNAs, and the cell type chosen for in vitro experiment is circumscribed to mimic the exact biological process during wound healing. Also, the novelty of this work is relatively limited since similar results using similar samples have been reached before.

*Reviewer #1 (Recommendations for the authors):*

Limited human-derived specimens and insufficient identification of clinical relevant miRNA hindered development of miRNA based therapy in venous ulcer treatment, this study performed an innovative work to acquire sufficient high-quality miRNA profiles from self-collected human skin specimens, and comprehensively analyzed miRNAs which participate in venous ulcer pathogenesis, and resolved the dynamic change within temporal dimension. The brief experimental validation also makes the conclusions more solid and reliable. While the data presentation form needs to be improved.

The project is ambitious and of interest to the medical community in dermatology as the compendium access is free of charge for academic institutions. Moreover, a certain validation of the data has been performed for the identified pathologically relevant miRs in abnormal VU expression.

1. The article could be updated on the presentation of clinical data or reference for miRs already found for VU or related dermatological disorders.

2. My main criticism is that the compendium database linked to the article is rather difficult to use for clinicians. I would recommend that the authors could propose a virtual visit with an example of miRs and mRNA expression etc, how to look for its level of expression in VU as well as their therapeutic indications.

3. Another criticism is that as mentioned in the article, extensive validation should be performed in cell type-specific miR expression data to confirm their findings. Otherwise, it may be difficult to make conclusions based on mixed cell types in biopsies from VU patients.

*Reviewer #2 (Recommendations for the authors):*

Liu et al. used biopsies from acute human excisional wounds and from chronic venous ulcers to identify miRNAs and mRNAs that are dysregulated in the ulcer tissue. Bioinformatics analysis of the data identified putative targets of strongly regulated miRNAs, which are also dysregulated in the ulcers. Finally, the authors show that some of the dysregulated miRNAs act cooperatively in keratinocytes to suppress their proliferation and to induce the expression of pro-inflammatory cytokines.

The combined analysis of acute and chronic human wounds is a strength. Such biopsies are very difficult to get and therefore, it is particularly appreciated that the authors developed a compendium that will be open to the public. The data analysis is very well done and the results are interesting and important for the field. However, the functional validation of the data is rather limited. There is no information on the cell type that expresses the relevant miRNAs and their targets and no spatial information. Functional studies are limited to keratinocyte proliferation (which is not a major problem in chronic ulcers). Therefore, the data remain speculative and the claims are not fully supported by the available data.

1. As mentioned by the authors, the observed changes may well be a consequence of a different cellular composition of acute vs. chronic human wounds. scRNA-seq would be the best way to address this issue, and this has now been done for many complex tissues. While such a complex analysis can of course not be done in a revision, the authors should at least make an effort to localize the most relevant miRNAs identified in this study and their putative targets in the wound tissue by in situ hybridization or immunostaining (for the targets).

2. Along the same line: The authors suggest a few transcription factors that could regulate the differentially expressed miRNAs (based on their bioinformatics analysis). To strengthen the data, the authors should perform immunostaining for these transcription factors to determine where they are expressed in acute vs. chronic wounds.

3. The functional data are limited to the analysis of the effect of two miRNAs in keratinocyte proliferation. However, it is not clear if these miRNAs are indeed mainly expressed (and regulated) in keratinocytes. In addition, keratinocyte proliferation is not a major problem in chronic wounds – in fact, keratinocytes at the wound edge of chronic ulcers are frequently hyperproliferative (Usui et al., 2008; Eming et al., 2014). The major problem is the impaired migration (Usui et al., 2008). Therefore, the authors should check if some of these miRNAs, which are abnormally expressed in VUs, are expressed in keratinocytes at the wound edge and if they regulate keratinocyte migration in vitro. Finally, the functional data were all obtained after overexpression of miRNAs, while knock-down studies are missing. Even if there is functional redundancy among the miRNAs of interest, it would still be useful to confirm this and to know if the endogenous levels are sufficient to contribute to a certain cellular function.

*Reviewer #3 (Recommendations for the authors):*

In this study, Zhuang Liu and colleagues performed paired-RNA sequencing in human skin tissues, including matched acute wounds and chronic non-healing venous ulcers. They identified 17 pathologically relevant miRNAs that exhibited abnormal VU expression and used qPCR to identify some miRNAs. The authors tried to find some targets which have the potential to accelerate wound healing, and this study has done good work on comprehensive analyses, the results are likely to be of interest to the scientific community. However, I have some serious concerns about this research.

1. The scientific novelty is limited, several groups have done similar analyses using similar samples before. Also, the sample size is limited in this study so the statistical power is weak to make solid conclusions.

2. The conclusion is rather weak because there are no experiments to prove the candidate miRNAs have the function in wound healing at all. Ki67 staining and live-cell imaging only suggest that miR-34a and miR-424 affect keratinocyte cell proliferation.

3. The paper is poorly written. Many regulatory network analyses have been done but those results were not clearly explained and it is hard to follow how conclusions were made based on such results.

4. Authors need to perform in vivo or in vitro assays to identify miRNA function.

5. Authors need to analyse previous data to confirm their findings.

---

## [Author Response]

[Editors’ note: the authors resubmitted a revised version of the paper for consideration. What follows is the authors’ response to the first round of review.]

Reviewer #1 (Recommendations for the authors):Limited human-derived specimens and insufficient identification of clinical relevant miRNA hindered development of miRNA based therapy in venous ulcer treatment, this study performed an innovative work to acquire sufficient high-quality miRNA profiles from self-collected human skin specimens, and comprehensively analyzed miRNAs which participate in venous ulcer pathogenesis, and resolved the dynamic change within temporal dimension. The brief experimental validation also makes the conclusions more solid and reliable. While the data presentation form needs to be improved.The project is ambitious and of interest to the medical community in dermatology as the compendium access is free of charge for academic institutions. Moreover, a certain validation of the data has been performed for the identified pathologically relevant miRs in abnormal VU expression.1. The article could be updated on the presentation of clinical data or reference for miRs already found for VU or related dermatological disorders.

As suggested by the reviewer, we have added a new Table S3 summarizing the reported miRNAs regulating skin wound healing, analyzed their expression pattern in our datasets (new Figure 10), and discussed the potential link between their expression and function patterns (the 4th paragraph in Discussion section). We show that most of the miRNAs that reportedly regulate wound healing also surface in our study, supporting the robustness of our profiling data and bioinformatics analysis.

2. My main criticism is that the compendium database linked to the article is rather difficult to use for clinicians. I would recommend that the authors could propose a virtual visit with an example of miRs and mRNA expression etc, how to look for its level of expression in VU as well as their therapeutic indications.

We thank the reviewer for this suggestion. We have added detailed virtual and text tutorials to our resource website to make it more user-friendly. Please visit the updated website: http://130.229.28.87/shiny/miRNA_Xulab/.

3. Another criticism is that as mentioned in the article, extensive validation should be performed in cell type-specific miR expression data to confirm their findings. Otherwise, it may be difficult to make conclusions based on mixed cell types in biopsies from VU patients.

We agree with Reviewer that it is crucial to validate the *in-silico* findings in cell type-specific miR expression data. During revision, we have dedicated major efforts to this aspect (new Figure 5a):

We purified epidermal keratinocytes from matched human skin and acute wounds of five healthy donors and then performed paired small RNA and long RNA sequencing in these purified cells. From these new experients, we obtained miR and mRNA in vivo expression profiles in keratinocytes of human skin and wounds. In particular, we confirmed the keratinocyte expression of the VU-related miRs and their targets (new Figure 5k and Figure 7a, b middle panels). In addition, with laser capture microdissection (LCM), we isolated epidermal compartments from human skin, acute wounds, and venous ulcers (n=7 per group) and confirmed the VU-related upregulation of hsa-miR-34a-5p and hsa-miR-424-5p occurred in wound-edge keratinocytes by qRT-PCR (new Figure 5l–n). These new data provide a rationale for selecting human primary keratinocytes for further functional validation. Next, we validated the miR-mediated gene regulatory network in human primary keratinocytes or fibroblasts for eight miRs by overexpressing each of them and performing microarray analysis with 33 samples (new Figure 6). By comparing these in vitro identified miR targets with the VU-in vivo gene expression signature, we identified the most pathological-relevant miR targets (new Figure 7) and the related signalling (Figure 8). We then examined the impacts of hsa-miR-34a-5p, hsa-miR-424-5p, and miR-516-5p and their combinations on the major biological processes of keratinocytes during human skin wound healing, i.e., inflammatory response, proliferation, and migration, using both gain- and loss-of-function approaches (new Figure 9). Although a complete picture of miR expression and function in each wound cell type is yet to be generated, our validation work focusing on keratinocytes strongly supports the robustness of our profiling data and bioinformatics analysis.

Reviewer #2 (Recommendations for the authors):Liu et al. used biopsies from acute human excisional wounds and from chronic venous ulcers to identify miRNAs and mRNAs that are dysregulated in the ulcer tissue. Bioinformatics analysis of the data identified putative targets of strongly regulated miRNAs, which are also dysregulated in the ulcers. Finally, the authors show that some of the dysregulated miRNAs act cooperatively in keratinocytes to suppress their proliferation and to induce the expression of pro-inflammatory cytokines.The combined analysis of acute and chronic human wounds is a strength. Such biopsies are very difficult to get and therefore, it is particularly appreciated that the authors developed a compendium that will be open to the public. The data analysis is very well done and the results are interesting and important for the field. However, the functional validation of the data is rather limited. There is no information on the cell type that expresses the relevant miRNAs and their targets and no spatial information. Functional studies are limited to keratinocyte proliferation (which is not a major problem in chronic ulcers). Therefore, the data remain speculative and the claims are not fully supported by the available data.

We appreciate the reviewer's positive comments and constructive suggestions for our work. We agree that the validation of the *in-silico* findings is crucial, and we have dedicated significant efforts to strengthen this aspect during revision (new Figure 5a):

We purified epidermal keratinocytes from matched human skin and acute wounds of five healthy donors and then performed paired small RNA and long RNA sequencing in these purified cells. With this approach, we obtained miR and mRNA in vivo expression profiles in keratinocytes of human skin and wounds. Particularly, we confirmed the keratinocyte expression of the VU-related miRs and their targets (new Figure 5k and Figure 7a, b middle panel). In addition, with laser capture microdissection, we isolated epidermis from human skin, acute wounds, and venous ulcers (n=7 per group) and confirmed the VU-related upregulation of hsa-miR-34a-5p and hsa-miR-4245p occurred in wound-edge keratinocytes by qRT-PCR analysis (new Figure 5l–n). These new data provide a rationale for selecting human primary keratinocytes for further functional validation. Next, we validated the miR-mediated gene regulatory network in human primary keratinocytes or fibroblasts for eight miRs by overexpressing each of them and performing microarray analysis with 33 samples (Figure 6). By comparing these in vitro identified miR targets with the VU-in vivo gene expression signature – we identified the most pathological-relevant miR targets (new Figure 7) and the related signalling (Figure 8). We then examined the impacts of hsa-miR-34a-5p, hsa-miR-4245p, miR-516-5p and their combinations on the major biological processes of keratinocytes during human skin wound healing, i.e., inflammatory response, proliferation, and migration, using both gain- and loss-of-function approaches (new Figure 9). Although a complete picture of miR expression and function in each wound cell type is yet to be generated, our validation work focusing on keratinocytes strongly supported the robustness of our profiling data and bioinformatics analysis.

1. As mentioned by the authors, the observed changes may well be a consequence of a different cellular composition of acute vs. chronic human wounds. scRNA-seq would be the best way to address this issue, and this has now been done for many complex tissues. While such a complex analysis can of course not be done in a revision, the authors should at least make an effort to localize the most relevant miRNAs identified in this study and their putative targets in the wound tissue by in situ hybridization or immunostaining (for the targets).

scRNA-seq has been used to analyze human skin and wound tissues (e.g., in our recent publication^1^); however, it remains challenging to analyze miRNAs at the single-cell level. In the few published protocols for single-cell small RNA sequencing, extensive cell handing is needed; therefore, this technology has only been applied to a few cells and is not ready to be used at a scale for analyzing complex dynamics of tissue^2^. Therefore, bulk small RNA-seq is still a suitable approach for miRNA profiling currently, and our results of analyzing the unique clinical samples will facilitate the research of miRNAs in human skin wound healing.

We agree that it is important to localize the miRNAs and their putative targets in the wound tissues, to exclude the possibility of cellular heterogeneity-related changes at least for some miRs identified in our study. As mentioned above, we have validated the in vivo expression of miRNAs and their putative targets in keratinocytes (Figure 5k and Figure 7a, b middle panel) and epidermis (Figure 5l–n) isolated from human skin and wound tissues by RNA-seq and qRT-PCR. These approaches allow us to detect many miRNAs and their putative target mRNAs simultaneously and gain an overview of the miR-mediated gene regulation in wound-edge keratinocytes. Also, we have localized one of the most relevant miRNAs identified in this study (i.e., miR-34a, Figure 5l, m) and its putative target LGR4 expression (Author response image 1) to wound-edge epidermal keratinocytes in human acute wounds and VU tissues in our recent work^3^.

**Author response image 1. sa2fig1:** Immunostaining of LGR4 in the skin, normal wound (NW), and VUs (n = 3). Adapted from our recent publication PMID: 31376385.

2. Along the same line: The authors suggest a few transcription factors that could regulate the differentially expressed miRNAs (based on their bioinformatics analysis). To strengthen the data, the authors should perform immunostaining for these transcription factors to determine where they are expressed in acute vs. chronic wounds.

We have performed immunofluorescence staining of GATA3 in our previous study (Figure 1B in Stojadinovic O, *et al.^4^*) and shown that GATA3 expression was detected throughout the epidermis of healthy skin but was completely absent in all VUs tested, confirming marked suppression of the main regulators of stem cell niche quiescence in non-healing edges of VUs^4^. KLF4 has been known for its crucial role in establishing skin barrier function^5^, and we have shown the downregulation of KLF4 expression in human VUs by microarray analysis in our previous study with an independent patient cohort^6^. As suggested by the reviewer, we are currently performing the immunostaining of KLF4 in human wound tissues, which will be completed for the revised version of the manuscript.

3. The functional data are limited to the analysis of the effect of two miRNAs in keratinocyte proliferation. However, it is not clear if these miRNAs are indeed mainly expressed (and regulated) in keratinocytes.

In the revised manuscript, by isolating keratinocytes (new Figure 5k) and epidermal compartments (new Figure 5l–n) from human wound tissues, we confirmed the expression and change of hsamiR-34a-5p and hsa-miR-424-5p in keratinocytes. We then examined the impacts of hsa-miR-34a5p and hsa-miR-424-5p on the major biological processes of keratinocytes during human skin wound healing, i.e., migration, inflammatory response, and proliferation (new Figure 5a and Figure 9).

In addition, keratinocyte proliferation is not a major problem in chronic wounds – in fact, keratinocytes at the wound edge of chronic ulcers are frequently hyperproliferative (Usui et al., 2008; Eming et al., 2014). The major problem is the impaired migration (Usui et al., 2008). Therefore, the authors should check if some of these miRNAs, which are abnormally expressed in VUs, are expressed in keratinocytes at the wound edge and if they regulate keratinocyte migration in vitro.

As suggested by the Reviewer, we have performed scratch wound healing assays, showing that simultaneous overexpression of miR-34a-5p and miR-424-5p delayed keratinocyte migration, whereas inhibition of endogenous miR-34a-5p or miR-424-5p enhanced keratinocyte motility (new Figure 9d–g). These new data suggest that miR-34a-5p and miR-424-5p, two miRs upregulated at the VU wound-edge keratinocytes (new Figure 5l-n), suppress keratinocyte migration, contributing to VU pathology.

Finally, the functional data were all obtained after overexpression of miRNAs, while knock-down studies are missing. Even if there is functional redundancy among the miRNAs of interest, it would still be useful to confirm this and to know if the endogenous levels are sufficient to contribute to a certain cellular function.

Besides overexpression of miRNAs, we also knock-down hsa-miR-34a-5p and hsa-miR-424-5p expression by transfection of miR-specific inhibitor to human primary keratinocytes in the revised manuscript (Figure 9f, g). We found that inhibition of either hsa-miR-34a-5p or hsa-miR-424-5p enhances keratinocyte migration, suggesting that their functions are not redundant.

Reviewer #3 (Recommendations for the authors):In this study, Zhuang Liu and colleagues performed paired-RNA sequencing in human skin tissues, including matched acute wounds and chronic non-healing venous ulcers. They identified 17 pathologically relevant miRNAs that exhibited abnormal VU expression and used qPCR to identify some miRNAs. The authors tried to find some targets which have the potential to accelerate wound healing, and this study has done good work on comprehensive analyses, the results are likely to be of interest to the scientific community. However, I have some serious concerns about this research.1. The scientific novelty is limited, several groups have done similar analyses using similar samples before. Also, the sample size is limited in this study so the statistical power is weak to make solid conclusions.

We appreciate the constructive suggestions from the Reviewer. Although the strategy of paired miRNA and mRNA analysis has been used in other diseases (e.g., cancers and neurodevelopmental disorders), to the best of our knowledge, our study is the first attempt to apply this strategy to human wounds. We have performed a literature search on PubMed and summarized all the published microRNA profiling data of human skin wounds (Author response table 1), which also supports the novelty of our study.

**Author response table 1. sa2table1:** Summary of published microRNA profiling data of human skin wounds.

References	Study design
J Clin Invest. 2015 Aug 3;125(8):3008-26.a study from our group	Skin from healthy donors (n = 5) at 0 hours and 24 hours after injury.
Burns. 2012 Jun;38(4):534-40.	Denatured dermis and paired normal skin of burn patients (n=3).
J Cell Physiol. 2022 Feb;237(2):1429-1439	Skin and day 14 wounds from both nonlesional and lesional sites from patients with vitiligo (n = 5).
Exp Dermatol. 2021 Aug;30(8):1065-1072.doi: 10.1111/exd.14405.	The parental control vs. iPSC-derived fibroblasts from non-healing diabetic foot ulcers, the intact foot skin of diabetic patients, or the healthy foot skin of non-diabetic patients (n = 2 or 3).

Our study has at least three strengths that enhance its novelty: (1) The human in vivo wound healing model allows us to monitor the in vivo dynamics of gene expression in human wound tissues and epidermal cells across the healing process. (2) The acute wounds were matched with venous ulcer in donor age and wound site – which has been rarely achieved in previous studies, although both aging and body locations are important factors affecting gene expression. (3) Paired small and long RNA-seq allows more accurate integrative multi-omics analysis. Our work leads to a novel and easy-to-use resource that can be used to prioritize clinical-relevant miRNAs for more in-depth functional studies and to evaluate the clinical relevance of previous findings of miRNAs studied in non-human models – such efforts are highly demanded in research community.

Regarding the sample size, this is a common challenge in the studies with human wound tissues^7^. Pastar I*. et al.* have summarized published studies using human wound tissues and stated each study's sample size (n=3-30 donors, please see Table 1 in the review article^7^). Pastar I*. et al.* commented that 'Traditional clinical trials that are testing efficacy typically involve hundreds of patients. Most clinical research sample size collections are markedly smaller, which often challenges their value. Therefore, one has to make a fundamental distinction between clinical trial and clinical research that is not just semantic. As described above, clinical research that focuses on deciphering molecular and cellular mechanisms that inhibit healing use comprehensive multipronged strategies involving modern technologies, and the type of research is often limited by rather moderate research funds and an allotted amount of time to complete it. Therefore, it is unrealistic to expect that the sample size of clinical research studies should be compared to that of clinical trials that are testing efficacy of therapies and interventions.' ^7^

After revision, we have analyzed tissue samples from 22 healthy donors (skin, day1, and/or day7 acute wounds were collected from each donor) and 19 venous ulcer patients in this study, which sample size is bigger than most existing studies using human wound tissues. Together with rigorous research design (e.g., skin and acute wounds from the same donor, matched VU and control patient cohort, paired small and long RNA-seq) and fortified efforts in experimental validation of the *in-silico* analysis (e.g., Figure 5–9), our study generated transcriptomic data of human wound tissues with relatively high robustness and reproducibility. Such efforts add a novel dimension to the current knowledge mostly relying on non-human models, and would be a valuable resource for further mechanistic studies of miRs with high translational potential.

2. The conclusion is rather weak because there are no experiments to prove the candidate miRNAs have the function in wound healing at all. Ki67 staining and live-cell imaging only suggest that miR-34a and miR-424 affect keratinocyte cell proliferation.

We agree with the reviewer that the functional validation of the identified miRs is a key, and we have dedicated major efforts to strengthen this aspect during the revision (Figure 5a):

In keratinocytes and epidermis isolated from human skin and wound tissues, we confirmed the keratinocyte expression of some VU-related miRs and their targets by RNA-seq and qPCR (Figure 5k–n and Figure 7a, b middle panel), which provide a rationale for the selection of human primary keratinocytes for functional validation. We next validated the miR-mediated gene regulatory network in keratinocytes or fibroblasts for 8 miRs by overexpressing each of them and performing microarray analysis with 33 samples (Figure 6). By comparing these in vitro identified miR targets with the VU-in vivo gene expression signature – we unravelled the most pathological-relevant miR targets (Figure 7) and their related biological processes (Figure 8). We then examined the impacts of hsa-miR-34a-5p, hsa-miR-424-5p, miR-516-5p and their combinations on the major biological processes of keratinocytes required for skin wound healing, including migration (scratch assays with live-cell imaging), proliferation (Ki67 qPCR, IF staining, and live-cell imaging), and inflammatory response (chemokine expression), using both gain- and loss-of-function approaches (Figure 9). Of note, many candidate miRNAs identified in this study have been shown to regulate wound healing in previous studies with in vitro and animal models (Table S3 and Figure 10), supporting the robustness and reproducibility of our profiling study. Together, our study provides an open resource for the research community that could evaluate the clinical relevance of previous findings of miRNAs studied in non-human models and prioritize the clinical-relevant miRNAs for more in-depth functional studies.

3. The paper is poorly written. Many regulatory network analyses have been done but those results were not clearly explained and it is hard to follow how conclusions were made based on such results.

Besides adding a significant amount of new experimental data (as mentioned above), we have substantively revised our manuscript by re-writing many parts of the text, as well as changing current figures and adding new figures and tables. We have also sent this manuscript to Enago for a professional language editing. We hope the updated version is clearer and easier to follow. We look forward to more suggestions to improve this manuscript further.

4. Authors need to perform in vivo or in vitro assays to identify miRNA function.

Please see our response to the 2^nd^ comment of Reviewer 3. In the updated manuscript, we have examined the impacts of hsa-miR-34a-5p, hsa-miR-424-5p, miR-516-5p and their combinations on the major biological processes of keratinocytes required for skin wound healing, including migration (scratch assays with live-cell imaging), proliferation (Ki67 qPCR, IF staining, and live-cell imaging), and inflammatory response (chemokine expression), using both gain- and loss-of-function approaches (new Figure 9).

5. Authors need to analyse previous data to confirm their findings.

In response to this excellent suggestion, we summarised the reported miRNAs regulating skin wound healing (Table S3), analyzed their expression pattern in our datasets (Figure 10), and discussed the potential link between their expression and function patterns (the 4^th^ paragraph in Discussion section). We found that most of the miRNAs that reportedly regulate wound healing also surfaced in our analysis, supporting the robustness and reproducibility of our profiling data and bioinformatics analysis.

References:

Li, D.*, et al.* Single-Cell Analysis Reveals Major Histocompatibility Complex IIExpressing Keratinocytes in Pressure Ulcers with Worse Healing Outcomes. *J Invest Dermatol* (2021).

Nielsen, M.M. & Pedersen, J.S. miRNA activity inferred from single cell mRNA expression. *bioRxiv*, 2020.2007.2014.202051 (2020).

Wu, J.*, et al.* MicroRNA-34 Family Enhances Wound Inflammation by Targeting LGR4. *J Invest Dermatol* 140, 465-476 e411 (2020).

Stojadinovic, O.*, et al.* Deregulation of epidermal stem cell niche contributes to pathogenesis of nonhealing venous ulcers. *Wound Repair Regen* 22, 220-227 (2014).

Segre, J.A., Bauer, C. & Fuchs, E. Klf4 is a transcription factor required for establishing the barrier function of the skin. *Nature Genetics* 22, 356-360 (1999).

Stojadinovic, O.*, et al.* Deregulation of keratinocyte differentiation and activation: a hallmark of venous ulcers. *J Cell Mol Med* 12, 2675-2690 (2008).

Pastar, I., Wong, L.L., Egger, A.N. & Tomic-Canic, M. Descriptive vs mechanistic scientific approach to study wound healing and its inhibition: Is there a value of translational research involving human subjects? *Exp Dermatol* 27, 551-562 (2018).